# Can Language Models Falsify?
# Evaluating Algorithmic Reasoning with Counterexample Creation

**Shiven Sinha**[1]    **Shashwat Goel**[2,3]    **Ponnurangam Kumaraguru**[1]
**Jonas Geiping**[2,3,4]    **Matthias Bethge**[4,5o]    **Ameya Prabhu**[4,5o]

[1]IIIT Hyderabad    [2] ELLIS Institute Tübingen
[3] Max Planck Institute for Intelligent Systems
[4] Tübingen AI Center    [5] University of Tübingen

## Abstract

There is growing excitement about the potential of Language Models (LMs) to accelerate scientific discovery. *Falsifying* hypotheses is key to scientific progress, as it allows claims to be iteratively refined over time. This process requires significant researcher effort, reasoning, and ingenuity. Yet current benchmarks for LMs predominantly assess their ability to generate solutions rather than challenge them. We advocate for developing benchmarks that evaluate this inverse capability — creating counterexamples for subtly incorrect solutions. To demonstrate this approach, we start with the domain of algorithmic problem solving, where counterexamples can be evaluated automatically using code execution. Specifically, we introduce REFUTE, a dynamically updating benchmark that includes recent problems and incorrect submissions from programming competitions, where human experts successfully identified counterexamples. Our analysis finds that the best reasoning agents, even OpenAI o3-mini (high) with code execution feedback, can create counterexamples for only $< 9\%$ of incorrect solutions in REFUTE, even though ratings indicate its ability to solve up to 48% of these problems from scratch. We hope our work spurs progress in evaluating and enhancing LMs' ability to falsify incorrect solutions — a capability that is crucial for both accelerating research and making models self-improve through reliable reflective reasoning.

## 1   Introduction

Empirical science has evolved through an iterative process of new claims, falsification and subsequent refinement of these claims. Mathematicians too follow a similar approach – they propose conjectures and then invest substantial effort in search of counterexamples before attempting a proof. There has been growing interest in using language models (LMs) to accelerate research (Jumper et al., 2021; Lu et al., 2024), which is considered the next frontier for AI progress. Yet, most existing LM benchmarks focus on a model's ability to *generate solutions* to problems (Jimenez et al., 2024; Wijk et al., 2024; Phan et al., 2025). In this work, we highlight the need for benchmarks that test the inverse capability – *falsification*. Scientific hypotheses are considered falsified when concrete contrary evidence is presented (Popper, 2005). Researchers create counterexamples to claims by inspecting every step of argumentation and evidence, leveraging domain knowledge, epistemic uncertainty, reasoning and creative intuition. This process requires deep understanding of the problem, and is both challenging and time intensive. Thus, benchmarking counterexample creation can not only accelerate scientific discovery, but also rigorously test LM reasoning abilities.

However, a key challenge with creating such benchmarks is to verify whether a model's output is a valid counterexample to the claim. Traditional benchmarking by comparison with ground-truth reference solutions is insufficient — there could be many valid counterexamples to a claim (Lakatos, 2015). In this paper, we make initial progress in this direction,

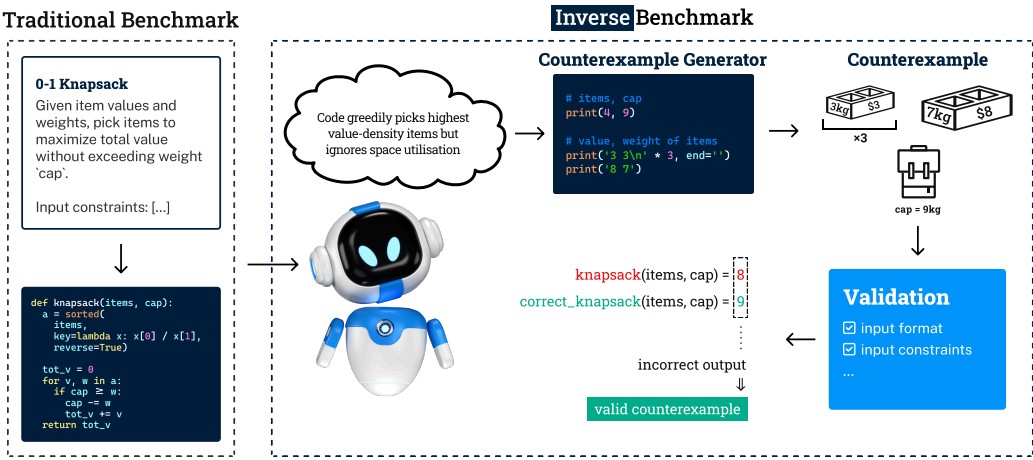

Figure 1: While standard benchmarks for algorithmic reasoning require models to generate solutions, we propose an inverse benchmark to evaluate reasoning about correctness by falsifying incorrect solutions. To allow expressivity, we let the model output a code that generates the counterexample input, and validate it by comparing the output of the incorrect solution with a held out correct solution.

by focusing on unstructured algorithmic problem solving, where counterexamples can be verified formally through code execution. Language models have already shown promise in solving algorithmic problems at an expert level (Jain et al., 2024; OpenAI, 2025). So we ask:

> Can LMs create counterexamples for
> incorrect solutions to algorithmic problems?

To investigate this, we design a novel benchmark: REFUTE (**R**efuting **E**rroneous **F**indings **U**sing **T**argeted **E**xamples). Each sample in REFUTE contains: (i) a detailed problem description (including input constraints) and (ii) an *incorrect* solution. The task for LMs is to create inputs satisfying the constraints, such that the given solution fails. The LM must generate a program that prints such an input, and is allowed to use arbitrary programmatic constructs to this end (Figure 1).

We automatically source samples of incorrect submissions from Codeforces* contests in 2024 and 2025. These samples are search and contamination free as Codeforces does not reveal non-trivial failure cases. We regularly update our benchmark to prevent indirect leakage due to potential overlap with training data, similar to LiveCodeBench (Jain et al., 2024). An automated pipeline to extensively filter samples ensures that counterexample discovery requires reasoning about the incorrect solution beyond random search. REFUTE contains 324 samples spanning diverse algorithmic topics with rich metadata annotations (Section 4).

We benchmark the current best models from five different model developers, as ranked based on their accuracy at generating solutions to these problems on LiveCodeBench (Jain et al., 2024). These include OpenAI o3-mini (high) and DeepSeek R1, which can generate correct solutions for up to 50% of the problems in our benchmark based on reported Elo ratings. Yet, even with few-shot prompting with chain of thought and a ReAct agent scaffold (Yao et al., 2022) with code execution feedback, these models can only find counterexamples to < 9% of problems in our dataset. Our results additionally demonstrate a prior hypothesis that LMs' ability to repair their own incorrect code is bottlenecked by their inability to find mistakes in their code (Gu et al., 2024; Olausson et al., 2024).

More broadly, our results demonstrate that verification, which includes falsification of subtly incorrect solutions, can sometimes be harder for models than solving the problem correctly. This indicates limitations in the potential for self-improvement using the generator-verifier gap (Song et al., 2024). As models progress towards solving novel problems where humans cannot provide ground-truth, it will be crucial that they can reflect about their

---

*https://codeforces.com/

own mistakes (DeepSeek-AI, 2025) to produce more reliable outputs. Our work takes first steps in evaluating a model's ability to falsify–in the limited domain of algorithmic problem solving, which is a form of deductive reasoning. We hope our work spurs interest in the community to create such "inverse benchmarks", that test the ability of models at invalidating incorrect solutions or claims in broader domains, including ones which involve inductive reasoning (Hume, 2000).

## 2  Related Work

**Formal verification** Programmatic search methods have long assisted researchers by enumerating and validating cases — ranging from symbolic execution tools like higher-order logic systems (Blanchette & Nipkow, 2010), SMT solvers (De Moura & Bjørner, 2008), symbolic execution tools like KLEE (Cadar et al., 2008) and randomized testing frameworks like QuickCheck (Claessen & Hughes, 2000) (for details see eloquent survey by Baldoni et al. (2018)). Automatic test case generation has widely been studied in software engineering (see surveys (Anand et al., 2013; Zhu et al., 1997; Runeson et al., 2006)). Existing work on LM based test generation (Li & Yuan, 2024) focuses on creating valid inputs and desired outputs for a given task specification. However, counterexamples may reside in non-obvious regions of combinatorially large input spaces, making search-based methods infeasible (Holzmann, 2002), especially in algorithmic reasoning settings (Forišek, 2006). While formal verification tools have made a lot of progress (Alur et al., 2013; Polgreen et al., 2020), their expressivity still remains limited (Ammons et al., 2002; Bjørner & Havelund, 2014; Rozier, 2016). Moreover, most claims are unstructured, and a complete formalization of the whole system is not possible for most claims. We focus on the task of hard-to-find, targeted test cases that invalidate subtly incorrect solutions.

**Fact Verification as Refuting False Information.** At first glance, fact checking (and misinformation detection) literature also aims to refute incorrect claims by verifying facts (Thorne et al., 2018; Guo et al., 2022; Press et al., 2024; Nakov et al., 2021), such as on social media platforms (Aïmeur et al., 2023; Chen & Shu, 2024) using sourced evidence (Nakano et al., 2021; Chen et al., 2023; De et al., 2024). Yet, fact checking is challenging due to unclear epistemological grounding of truth in a complex social world (Uscinski & Butler, 2013; Vinhas & Bastos, 2022). In contrast, we focus on domains with clear truth semantics (Davidson, 1967), requiring the model to produce a counterexample that *verifiably* refutes a given claim.

**Cybersecurity.** Cybersecurity has traditionally focused on vulnerability detection with CTF style contests (Agency, 2023; AIxCC, 2016), with recent interest in LLM Agent systems (Abramovich et al., 2024; Deng et al., 2023) (refer to (Motlagh et al., 2024) for a survey). Similarly, *fuzzing* (Huang et al., 2024b) involves finding security vulnerabilities in software codebases by providing invalid or unexpected random data and monitoring for crashes, failing pre-defined assertions, memory leaks. On the surface, CTF and fuzzing do require finding worst-case inputs that cause code to misbehave. Our work focuses on creating counterexamples using valid inputs for subtly wrong algorithmic solutions, with an eye towards subtly incorrect scientific claims.

**Code Self-Repair.** Existing work on code self repair (Chen et al., 2024) focuses on the following task: the LM is given access to a code execution environment (Wang et al., 2024) along with predefined test cases, and it iteratively improves its own solution using compilation and correctness feedback till it passes all test cases (Khattab et al., 2023). Instead, our work focuses on the ability of the LM to generate worst-case test cases where a given incorrect solution would fail. Gu et al. (2024) show that LMs struggle to classify their own generations as incorrect. Improving counterexample creation could thus help LMs self-repair their own code when solving novel problems with unknown ground-truth.

**Language Models for Scientific Discovery.** FunSearch (Romera-Paredes et al., 2024) demonstrated the use of LMs to generate novel solutions to an open problem in Mathematics, given access to a programmatic evaluator. We ask, can LMs falsify subtly incorrect solutions or claims, including domains where programmatic evaluation might not be possible? Lu et al. (2024); Si et al. (2024) proposed writing research papers end-to-end using an LM, an agentic pipeline where feedback is obtained from LM generated review scores. Instead of

depending on papers and review scores as arbitrary units of science, we focus on research progress through precise claims and counterexamples.

**Scalable Oversight.** In the limit, improving at counterexample creation can help models red-team (Perez et al., 2022) another model's reasoning (Tyen et al., 2024; Huang et al., 2024a), thus acting as better judges (Zheng et al., 2023). Allowing self-improvement through critique (Wang et al., 2025) and debate (Kenton et al., 2024). Finding mistakes and avoiding sycophantic behavior (Sharma et al., 2024) is crucial for emerging safety paradigms where models assist humans in overseeing other models (Bowman et al., 2022).

# 3  Problem Formulation

Most current benchmarks for language models involve choosing or generating a correct solution to a given problem. We now formalize inverse benchmarks, which test a model's ability to falsify incorrect solutions.

## 3.1  Falsification by Providing Counterexamples

To falsify a *claim* means to find *a counterexample* that shows the claim is not always true.

- *Claim*: A claim $\mathbb{C}$ has two parts: a set of conditions $\mathcal{H}$ and a proposition $\mathcal{P}$. The claim is true when: Given any input $x$ that meets the conditions ($\mathcal{H}(x)$ is true), then it must also make the proposition true ($\mathcal{P}(x)$ is true). We write this as $\mathcal{H}(x) \implies \mathcal{P}(x)$.
- *Counterexample*: A counterexample to a claim $\mathbb{C}$ is an input $x^*$ that shows the claim is wrong. This means that $x^*$ follows all the rules ($\mathcal{H}(x^*)$ is true) but does not make the statement true ($\mathcal{P}(x^*)$ is false).

**Task**. Given a claim $\mathbb{C}$ (which might be stated in natural language), the model must find a counterexample $x^*$.

## 3.2  Finding Counterexamples for Algorithms

How to check whether a proposed counterexample $x^*$ truly invalidates the claim? This requires verifying: (i) The conditions $\mathcal{H}$ are met and, (ii) The claim $\mathcal{P}$ does not hold for the given input. As a first step, we focus on a setting where counterexamples can be automatically verified: algorithmic problem solving. Here, a solution is to be generated for a problem statement which specifies the computational task, input constraints and input-output formats, and example cases for reference. The conditions $\mathcal{H}$ and claim $\mathcal{P}$ are:

$\mathcal{H}$ : The input format and constraints included in the given problem statement are satisfied.

$\mathcal{P}$ : The given code $\mathcal{A}$ solves the problem described in the statement.

**Task**. The goal of the model is to find an input $x^*$ where $\mathcal{A}$ produces an incorrect output. A validator script verifies whether $x^*$ satisfies the input constraints $\mathcal{H}$. Then, the claim can be checked by comparing the output of $\mathcal{A}$ to a ground-truth solution ($\mathcal{A}^*$), i.e., $\mathcal{A}(x^*) \neq \mathcal{A}^*(x^*)$.

# 4  REFUTE Benchmark

Having formalized counterexample creation in algorithmic problem solving, we now describe the data collection process and features of REFUTE.

## 4.1  Data Collection

We first discuss how we collect problems and corresponding incorrect codes that form the samples in REFUTE. The overall pipeline is described in Figure 2.

**Sourcing Problem Statements**. We first source 647 problems from Codeforces Division 1 and Division 2 contests held between January 2024 and January 2025, reflecting latest,

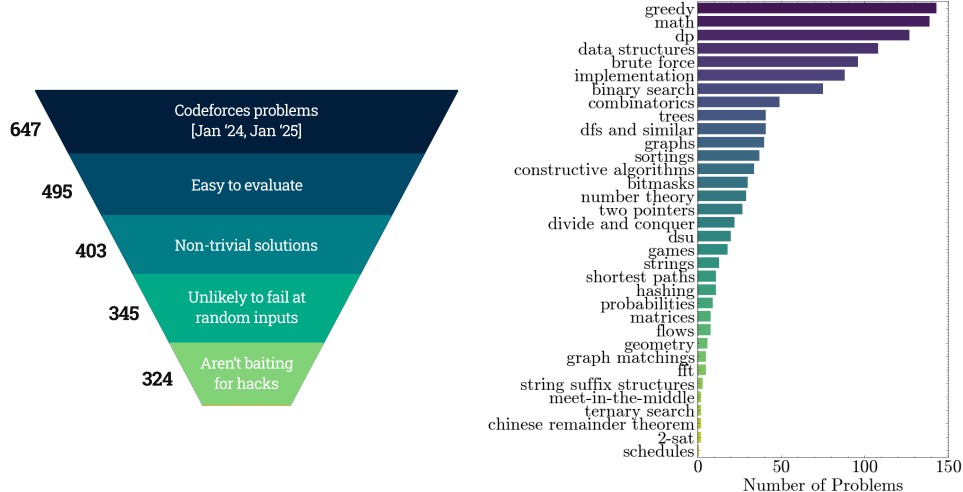

Figure 2: **(Left) Sample Filtering Pipeline.** We curated (problem, incorrect solution) pairs from Codeforces (Jan 2024–Jan 2025) where counterexamples are easy to verify but non-trivial to generate, using a 4-step filtering process (dynamically expandable over time). **(Right) Dataset Topic Coverage.** The dataset covers 35+ algorithmic topics, including many niche ones. Problems are multi-tagged—e.g., the high number of "greedy" problems indicates it's one of the key concepts, not the only one.

competition-level programming tasks. We apply two filters on the problems: (i) We remove 152 (23%) problems that require a non-trivial grading environment for any given input. Such problems are explicitly marked as involving interaction between the program and a grader, or allowing multiple correct outputs for a given input. (ii) Then, we remove 92 (18% of the remaining) problems which are rated below 1200. The incorrect solutions here are more likely to have trivial implementation bugs as these problems do not require much algorithmic reasoning. We also obtain the correct solution for each problem as these are released officially by the problem authors in an editorial.

**Picking Incorrect Solutions**. Next, we select incorrect submissions for each of the shortlisted problems, optimizing for ones where creating counterexamples might be more challenging and interesting. As we wish to test falsification and not implementation inefficiences, we filter to submissions marked as providing wrong answers, rather than those that violated time or memory constraints. For these incorrect submissions, we prioritize them using the following scoring function: $\text{score}(s) = h(s) \cdot M + t(s) + 10 \cdot b(s)$. Here, $h(s)$ indicates whether the submission was "hacked" post-contest after passing all test-cases designed by the problem authors. We strictly prioritize such submissions by adding a large constant $M = 10^4$ to their score, as they form interesting instances of subtle incorrect solutions that slipped through initial tests, but human experts could create counterexamples for them under time-constraints. Next, $t(s)$ is the number of test cases the submission passed before giving a wrong answer. Finally, $b(s)$ takes a binary value that indicates whether the author is rated $\geq 2000$, acting as a bonus for expert written incorrect submissions.

**Filtering Trivial Samples**. Finally, we remove samples for which finding counterexamples is trivial in two ways. First, we wish to ensure that randomly generating test cases without reasoning about the incorrect code is not enough to find a counterexample. We provide Gemini 2.0 Flash Thinking just the problem statement, without the incorrect code, and prompt it to output a random test case generation code. We run this test case generator for up to one minute to check whether it can find any inputs where the incorrect code doesn't match the ground-truth solution. This is true for 58 (14%) of the 403 problems. We filter these incorrect submissions.

Second, we found that 21 (6%) submissions contain deliberately inserted code that produces incorrect outputs only when a specific, unlikely constant appears in the input. This is done perhaps as a way to maliciously bait for hacks in the contest, as hackers are rewarded with extra scores. For the results in our paper, an expert human evaluator went through the submissions and removed such samples. It is easy to automate this step as the malicious

Figure 3: **(Left)** Distribution of Test Cases Passed by Incorrect Solutions: The median test cases passed is 17, making the mistakes non-trivial. **(Center)** Distribution of Problem Ratings: The benchmark spans a range of difficulty levels. **(Right)** Distribution of Incorrect Solution Author Ratings: Preference was given to expert authors, rated above 2000, resulting in a noticeable peak.

parts of such code are quite overt. We plan to utilise a language model by providing it the expert identified samples as demos, along with a rubric.

**Final Dataset**. The REFUTE dataset contains 324 samples authored by 304 different programmers. The incorrect submission in 317 samples is written in C++. The remaining 7 are in Python, as C++ is far more popular in programming competitions due to its efficiency. Each sample consists of a unique problem statement with an incorrect code solution, and the correct solution along with an input validation script is available for evaluation. The corresponding lengths are summarized in Table 5.

## 4.2 Benchmark Features

Our benchmark is constructed to have desirable features highlighted below.

**Allows arbitrary algorithmic generation of novel counterexamples**. We provide the language model a problem statement and incorrect solution, and ask it to output a code $\mathcal{A}_{out}$ that, when executed, outputs a counterexample input $x$. The code $\mathcal{A}_{out}$ must complete its execution within 1 minute, and the LM is informed of this time limit in its prompt. Allowing the LM to output code allows it to generate counterexamples with varying complexity, ranging from a hard-coded input to complex functions that create the counterexample input. We score the counterexample as a success if the incorrect solution has a different output from the correct one, while matching constraints specified in the problem.

**Avoids Search and Training Data Leakage**. Codeforces does not publically reveal the full test cases that broke an incorrect submission on non-trivial cases, so models cannot directly find counterexamples on the internet. Further, to prevent indirect leakage from user discussions, we will dynamically update the claims as more contests on Codeforces occur, similar to LiveCodeBench (Jain et al., 2024). This allows model comparisons by filtering to the subset of claims collected after the latest knowledge cutoff date among the models.

**Diversity and Metadata**. Our benchmark spans 34 fundamental topics in algorithms as tagged by Codeforces (e.g. Greedy, Dynamic Programming, Graphs, etc.), shown in Figure 2. The problems range in difficulty from an Elo rating of 1200 to 3500, while the incorrect submissions are authored by programmers with expertise ranging from Elo 700 to 3800. Figure 3 shows the distribution over problem rating, solver rating, and the test case number where the submission failed on CodeForces. We provide all these meta-data annotations for each sample in our benchmark, which may be helpful for future research.

# 5 Experiments

We first report frontier reasoning and non-reasoning models' performance at creating counterexamples on our benchmark and then analyze the results.

## 5.1 Models struggle at creating counterexamples

Based on LiveCodeBench (leaderboard in Figure 5), we select the three best reasoning models from unique developers: o3-mini-high-0131, DeepSeek R1, Gemini 2.0 Flash Thinking-0121, and the two best chat models, Claude-3.5-Sonnet-20241022 and DeepSeek V3. We also test Claude Sonnet 3.7, enabling thinking in the zero-shot setting.

Table 1: **Performance Comparison of LLMs.** We test 4 strategies across 324 problems in our benchmark. While top reasoning models can generate correct solutions for nearly half of the problems, their ability to find counterexamples for subtly incorrect solutions lags significantly, even with agentic code execution feedback. Providing models with the correct solution does not substantially improve counterexample generation, highlighting a fundamental gap between solving and falsifying.

| Model | Solution Generation | | Counterexample Creation | | | | | |
|---|---|---|---|---|---|---|---|---|
| | Rating | Solved% | Cost ($) | Prompting | | | ReAct Agent | |
| | | | | Zero-shot | Few-shot | w/ Correct | w/o Demo | w/ Demo |
| DeepSeek-V3 | 1134 | 10.8 | 5 | 2.4±1.7 | 2.7±1.8 | 3.7±2.1 | 3.7±2.0 | 3.1±1.7 |
| Sonnet 3.5 | 717 | 6.6 | 71 | 4.6±1.9 | 3.7±2.0 | 2.2±1.6 | — | 3.0±1.9 |
| Sonnet 3.7 | – | – | 383 | 8.0±2.4 | – | 3.4±1.4 | – | 7.5±2.3 |
| Flash 2.0 (Thinking) | – | – | 0 | 0.9±1.0 | 2.1±1.6 | 2.5±1.7 | 1.8±1.4 | 2.5±1.7 |
| DeepSeek-R1 | 2029 | 44.0 | 104 | 5.8±2.5 | 5.2±2.4 | 4.6±2.1 | — | 6.5±2.6 |
| o3-mini (high) | **2130** | **48.7** | 76 | 8.6±2.8 | 8.9±2.9 | 9.3±2.8 | 6.8±2.6 | 8.6±2.8 |

We compare two strategies: *Prompting* and *Agentic Code Execution*.

**Prompting.** We prompt each model with the problem statement, input constraints, example input-output pairs, and the incorrect code. The model must produce a script to print a failing test case, along with concise reasoning. In the few-shot setup, we also present three sample problems and incorrect code illustrating diverse issues with expert-annotated rationales.

**Agentic Interaction with Code Execution Feedback.** A typical human workflow of finding bugs involves tinkering with the code and observing its behaviour on various inputs. Inspired by this, we allow the model to interact with a code execution environment. The model can make upto ten attempts to execute any code with arbitrary inputs it wishes to test. It receives the output in return. In case of errors, it receives the corresponding feedback instead (e.g. compiler messages). For example, the model can add print statements to the incorrect code and observe intermediate behaviour ([Hu et al., 2024]). It can also write its own versions of subroutines in the code and observe any differences from how the buggy code handles the same scenario, effectively avoiding the need to "dry run" computations itself. To prevent exhausting the context, we truncate the outputs to upto 2000 characters before revealing them to the model. Each code execution is limited to 30 seconds. After this interactive phase, similar to the standard prompting setup, the model must submit a script to print a failing test-case. If this submission fails validation checks, we provide this feedback to the agent and allow it to resubmit upto five times.

**Can models solve these problems?** We include the *code generation* performance of models on our benchmark's problems by estimating the number of problems solved. This is computed by first finding the per-problem success probability, which is derived from the problem's rating and the model's reported Elo ([DeepSeek-AI, 2025]) following standard Codeforces rating calculations. If the model is rated $r$, we define: $\text{solved}(r) = \text{E}_p \left[ \frac{1}{1+10^{\frac{p-r}{400}}} \right]$

Here, $p$ is sampled from the problem ratings in our benchmark.

**What if models have access to the ground-truth solution?** We also evaluate the impact of providing models with access to the correct code. This decouples the advantage of models with better solution generation abilities. We augment the earlier zero-shot prompting setup to additionally reveal the correct solution and report results in Table 1 *(w/ Correct)*.

**Discussion and Error Analysis.** Table 1 shows that counterexample creation lags significantly behind solution generation and does not scale proportionally. Models also struggle to leverage code execution feedback—a key component of human debugging workflows—with only DeepSeek R1 exhibiting modest improvements. On the other hand, our analysis shows that execution feedback greatly reduces test-case validation failures. For instance, DeepSeek R1 and V3 both eliminate validation issues completely, compared to 45 and 36 failures in the zero-shot setting. Additionally, while few-shot prompting with expert rationale improves Gemini's performance over zero-shot, other models show minimal gains or even degrade.

Table 2: **Prompting Models to use Search to find Counterexamples.** Without the correct solution (RandSearch), models often generate invalid counterexamples where the incorrect solution gives the right output. The counterexample success rate increases with access to the correct solution (RandSearch Oracle), but significant room for improvement remains.

| Model | DeepSeek-V3 | Flash 2.0 (Thinking) | DeepSeek-R1 | o3-mini (high) |
|---|---|---|---|---|
| RandSearch | 4.0 | 3.7 | 4.0 | 8.3 |
| RandSearch Oracle | 15.1 | 7.7 | 9.9 | – |

The oracle (Table 1 *(w/ Correct)*) simulates a hypothetical where o3 (currently unreleased) matches the reported Codeforces rating of 2727, or a future model is able to solve most Codeforces problems: would they automatically be able to find counterexamples for incorrect solutions? While counterexample creation abilities could also improve, knowing the correct solution alone is insufficient even for the best current reasoning model, o3-mini (high).

## 5.2 Does explicit prompting for search help?

Manual inspection of model outputs revealed that models rarely used a search based strategy to find counterexamples. In contrast, humans often generate randomized inputs guided by structural intuition which they expect to yield valid counterexamples. To address this, we explicitly prompt models to generate counterexamples using a search-based strategy with controlled randomization. Note that a key difference from our initial filtering step (where we filtered trivial samples that can be broken through search without reasoning, described in Section 4.1) is that there the model was not given access to the incorrect solution, so it by definition did not reason about it, whereas here it can. Specifically, we test two strategies and report these numbers in Table 2:

**RandSearch.** The model constructs a randomized input generator and a brute-force solution. We use the generator to search for tests and compare the outputs of the brute-force solution against the incorrect code. The search terminates when it encounters differing outputs. We limit this search to 2 minutes. We provide few-shot examples with rationale only to non-reasoning models, as reasoning models performed better without them.

**RandSearch Oracle.** The previous step requires the model to generate a brute-force solution. While it is often easy to find exponential-complexity solutions for these algorithmic problems, models could still produce wrong ones. To alleviate this, we simulate a hypothetical scenario where the model has access to the correct solution and then writes a randomized search strategy to find an input where the incorrect solution's output strategy doesn't match. In other words, the model can use our ground-truth verification environment, so any input produced at the end is likely to be correct as long as it passes validation checks.

We provide a detailed description of the motivation and nuances of these methods in Appendix A.

**Discussion and Error Analysis.** Performance of reasoning models deteriorates when explicitly prompted to use randomized search. It offers marginal gains for chat models. For o3-mini, the successful samples are interestingly quite disjoint when using this strategy. The counterexamples created invalidate 6% distinct submissions that the previous prompting and agentic strategies could not, in contrast to 3% new submissions invalidated when the model was provided correct solutions. This shows that models learning to leverage programmatic search when appropriate can significantly boost performance.

Further analysis of RandSearch's failure modes reveals that, on average, 35% of all samples terminate with an incorrect counterexample. Among these, 14% fail due to test-case validation errors, while the remaining 86% result from incorrect brute-force code. This trend remains remarkably consistent across models — differences in code generation benchmarks do not appear to significantly impact their ability to write trivial brute-force code.

Gemini Flash 2.0 (Thinking), the worst performer, and o3-mini (high), the best, are similarly bottlenecked by incorrect brute-force code (28% vs. 32% of all samples). However, o3-mini (high) still manages to double its overall success rate, suggesting that more intelligent search strategies can outperform unguided attempts.

Figure 4: **Analysis of R1, o3-mini (high) success rate at creating counterexamples, combining success across strategies like prompting, agent and programmatic search**. We find that model successes are not predictable based on problem difficulty, or attributes of the incorrect submission like author expertise and number of test cases passed before giving a wrong answer. Success rate on the hacked subset, where counterexamples were found despite passing all initial tests is 4% ±1%.

## 5.3 When can models create counterexamples?

As counterexample creation abilities are particularly relevant for model reflection and reliability in model outputs, it is important to characterize in what types of situations they can and cannot create valid counterexamples. We analyze counterexample creation success across three attributes which are highly predictive of solution generation correctness – problem difficulty, number of tests the incorrect submission passed before failing, and incorrect submission author expertise. For each attribute, we divide samples into five percentile buckets and measure average success at creating counterexamples of the best two models, R1 and o3-mini (high), across prompting (Zero-shot, Few-shot), agentic (with and without demos) and search strategies.

Surprisingly, we find no clear trends in Figure 4 between these three attributes and counterexample creation success. We also find limited insight based on problem statement and incorrect solution length as shown in Appendix Figure 6. This demonstrates that model counterexample capabilities can be non-trivial to predict using attributes that are predictive of solution generation capabilities. This further highlights the complementary nature of falsification. We believe that better understanding of what attributes contribute to its difficulty is an important direction for further investigation.

## 5.4 Where do the models fail?

To better understand the failure modes of the models, we conducted a detailed error analysis. A domain expert (Codeforces Grandmaster) manually annotated the underlying bugs in the incorrect code and compared them against the models' attempts at generating counterexamples.

### 5.4.1 Analysis of Bug Misclassification

In our analysis of 25 random tasks, we observed the following tendencies in how models classified the nature of the bug.

- For 48% of the tasks, the model incorrectly assumed the bug was with the underlying logical approach, completely missing that it was a minor implementation error.
- Conversely, for only 8% of the samples, the model mistakenly identified an issue as an implementation bug when the core logic was flawed.
- The model correctly identified the bug as a logical issue in 20% of the cases and as an implementation bug in 24% of the tasks.
- For 20% of the tasks, the model had a nearly correct understanding of the bug but was unable to generate a test case to exploit it.

For these same incorrect submissions, we examined the performance of the original human contestants in the programming competition. We found that the contestants took a median time of 4 minutes to identify the bug, correct it, and resubmit an accepted solution, indicating that for humans, these were often not conceptually difficult bugs to resolve.

Table 3: Distribution of successfully identified bug types compared to their overall distribution in a random sample of the dataset.

| Bug Type | % of Successful Tasks | % in Dataset (Sampled) |
|---|---|---|
| Initialization and bounds errors | 56% | 32% |
| Incorrect approach | 28% | 28% |
| Implementation errors | 12% | 20% |
| Edge case handling errors | 3% | 20% |

### 5.4.2 *Qualitative Analysis of Failures*

We performed a qualitative analysis of the failed attempts to identify recurring patterns of incorrect reasoning.

- **Misallocated Attention.** Models frequently misdirect their attention to parts of the code less likely to contain bugs, such as I/O code or standardized data structure templates. This contrasts with expert human programmers who typically begin by critiquing the high-level algorithmic logic before focusing on lower-level implementation details. This suggests that models may lack effective abstraction for evaluating the structural correctness of a program.

- **Boundary Condition Analysis.** Models frequently fail to find bugs related to edge cases. Human debuggers often test extremal values within the given input constraints to find such bugs. Despite many instances in our benchmark being susceptible to these types of errors, the models often failed to generate the corresponding counterexamples.

- **Failure to Follow Search-Based Instructions.** When prompted for randomized search code, reasoning models often ignored the instruction, instead outputting a single deterministic test case after a long chain of reasoning. In contrast, chat models such as DeepSeek V3 consistently followed the instructions, leading to better performance.

- **Impact of Feedback on Validation Errors.** A significant number of counterexamples, particularly from DeepSeek R1, were invalid due to format or constraint violations (almost 15% in zero-shot and few-shot settings). Providing explicit feedback (agentic setting) on why a test case failed proved highly effective and nearly eliminated these errors, whereas simple failure notifications were insufficient.

### 5.4.3 *Distribution of Correctly Identified Bugs*

We further analyzed all the successfully generated counterexamples across models and strategies to understand which types of bugs are more easily identified (Table 3). This distribution is notably skewed when compared to the underlying distribution of bug types in a random sample of 50 problems from our dataset, which is more balanced. Models are significantly better at finding simpler issues like off-by-one errors or the use of incorrect constants. Conversely, they are less likely to identify instances where the fundamental algorithmic approach is flawed, and they perform even worse when the logic is incorrect only in specific, corner cases that could be handled separately.

## 6 Conclusion

In this work, we take the first steps toward benchmarking the ability of language models to create counterexamples for incorrect solutions. Focusing on algorithmic problem solving, we found that models often fail to detect errors in failed Codeforces submissions — even when given the correct solutions as guidance. We hope hillclimbing on our benchmark spurs effective methods for creating counterexamples using LMs, and deepens our understanding of how this capability relates to a models capacity for reasoning, reflection, and self-improvement. We hope there will be more benchmarks for counterexample creation, such as for research-level mathematics (Bengio & Malkin, 2024; Davies et al., 2021; Wagner, 2021). We are excited about methods that integrate formal tools such as SMT solvers to leverage their complementary strengths. While our work focuses on claims and counterexamples verifiable through code execution — many scientific hypotheses are not easily formalized in this way. Designing evaluations that test a model's ability to propose counterexamples based solely on natural language claims presents an exciting direction for future research.

## Impact Statement

AI generated solutions, such as entire research papers (Lu et al., 2024), risk overwhelming our infrastructure to refute them, following Brandolini's law (Williamson, 2016). Our paper highlights the need for inverse benchmarks, that evaluate the ability of models to falsify subtly incorrect solutions, instead of the traditional paradigm of solving problems. Improved ability to falsify can help make AI outputs more reliable through reflection, which is important as edge-cases have long been the Achilles heel of deep learning driven deployments like self-driving. To emphasise that falsification can sometimes be harder than generation, we create REFUTE, which we will release publically. Progress on REFUTE could also help language models' ability to audit codebases for mistakes, improving software reliability in an increasingly digitized economy.

## Author Contributions

Shiven and Ameya conceived the project. Shiven led the experiments, and Shashwat helped design the benchmark. Shashwat and Shiven co-led the writing of the paper, with help from Ameya. PK provided helpful feedback throughout the project. Jonas, Matthias and Ameya advised the design of all experiments.

## Acknowledgements

Special thanks to Siddharth Bhat and Siddhartha Gadgil for inspiration about this direction and specifically for introducing us to formal verification and counterexample creation. The authors would like to thank (in alphabetical order): Nikhil Chandak, Hari Aakash K, Shyamgopal Karthik, Srija Mukhopadhyay, Alexander Panfilov, Vishaal Udandarao, Saujas Vaduguru. AP and MB acknowledge financial support by the Federal Ministry of Education and Research (BMBF), FKZ:01IS24085B and Open Philanthropy Foundation funded by the Good Ventures Foundation.

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

# Appendix

We now provide thorough details about the benchmark, baselines, and prompts.

## Contents

# A More Details About Baselines

## A.1 Random Search

This method is motivated by the observation that it is often trivial to solve algorithmic problems under smaller constraints, potentially allowing suboptimal efficiency. For instance, consider the task of finding a minimum spanning tree of a connected graph. A trivial solution is to try all possible subsets of edges. Among all such subsets, we pick one which retains the connectivity of the graph while minimising the sum of edge weights. This solution doesn't require any elaborate reasoning and follows directly from the definition of an MST. Notice that this is much slower than optimal solutions like Kruskal's algorithm (Kruskal, 1956), which run in $\mathcal{O}(E \log E)$ instead of our inefficient $\mathcal{O}(E2^E)$. However, this inefficient solution can execute within a few seconds for smaller graphs, with say around 20 edges, and provide valuable ground-truth outputs for arbitrary (small) inputs. This can be then used to check the correctness of the efficient but buggy solution. It is still possible for a model to write an incorrect solution or to make a test-case generator that prints invalid inputs to the problem. In such cases, the test-case discovered by this random testing will fail the subsequent evaluation. The model also needs to ensure that the random test-cases generated by its script are small enough that its own correct solution can finish execution in time.

We prompt the model to output two pieces of code: (1) a randomised test-case generator following the input constraints of the problem, and (2) an inefficient solution to the problem. This could be simple brute-force enumeration of all possibilites for small constraints, and finding the optimal solution using them. After obtaining these, we repeatedly generate test-cases using the generator until we find an input for which the answers provided by the incorrect and the model-generated solution differ. If such a case is found within 1 minute of execution time, it is taken to be the attempted counterexample and evaluated. Otherwise, this search for a test-case terminates with the model receiving no credit. We include three demonstrations for creating a randomised input generator along with a simple correct solution in our prompt.

## A.2 Random Search (Oracle)

We augment the Random Search baseline to reveal the efficient, correct solution to the model alongside the initial setup of providing the problem details and the buggy code. This has three important consequences.

1. **Comparative Analysis.** The model now knows the exact steps to solve an arbitrary instance of the problem. It does not have to engage in careful analysis and reasoning to arrive at this. The counterexample search can be guided by comparing the high-level conceptual approaches of the two implementations, as well as the low-level details such as handling of edge cases, array sizes, and variable bounds.

2. **Large Randomised Generations.** Unlike in RandSearch, where the correct code was prohibitively inefficient, the model's test-case generation script is no longer forced to output smaller test-cases. Both the correct and incorrect solutions can now run quickly under the full scale of the original problem's constraints. As a result, random testing is much more likely to uncover bugs that only arise with larger problem instances, e.g. overflows, out-of-bounds access, edge cases, etc.

3. **Eliminates Error in 'Assumed Truth'.** In RandSearch, the search for an input that caused a discrepancy between the two codes often terminated quickly. However, the issue was frequently an inaccurate brute-force solution generated by the model rather than a bug acting up in the provided incorrect code. In this new setup, if the search for a failing test-case terminates within 1 minute, it is guaranteed to pass the subsequent evaluation phase (given that it follows the input format).

# B  More Details About Evaluation

**Environment.** In line with the judging environment used by Codeforces, we perform our experiments on a Windows machine. We also mimic the compilation flags for C++, specifically: `-Wall -Wextra -Wconversion -static -DONLINE_JUDGE -Wl,-stack=268435456 -O2 -std=c++23 program.cpp -lstdc++exp`. For executing the incorrect and ground-truth code on inputs, we allow a liberal time limit of 30 seconds. This is more than the limit imposed by Codeforces for the problems in REFUTE, which is at most 15s with a mean of 2.4s.

**Programming Languages.** In REFUTE, we provide the exact programming language description as supplied by Codeforces. The distinct languages spanned by the dataset are: C++14 (GCC 6-32), C++17 (GCC 7-32), C++17 (GCC 9-64), C++20 (GCC 11-64), C++20 (GCC 13-64), C++23 (GCC 14-64, msys2), PyPy 3-64, and Python 3. For execution, all C++ programs are compiled with `-std=c++23`, leveraging backward compatibility We use the standard CPython interpreter. We verified that these choices do not alter the behavior of the code in our benchmark.

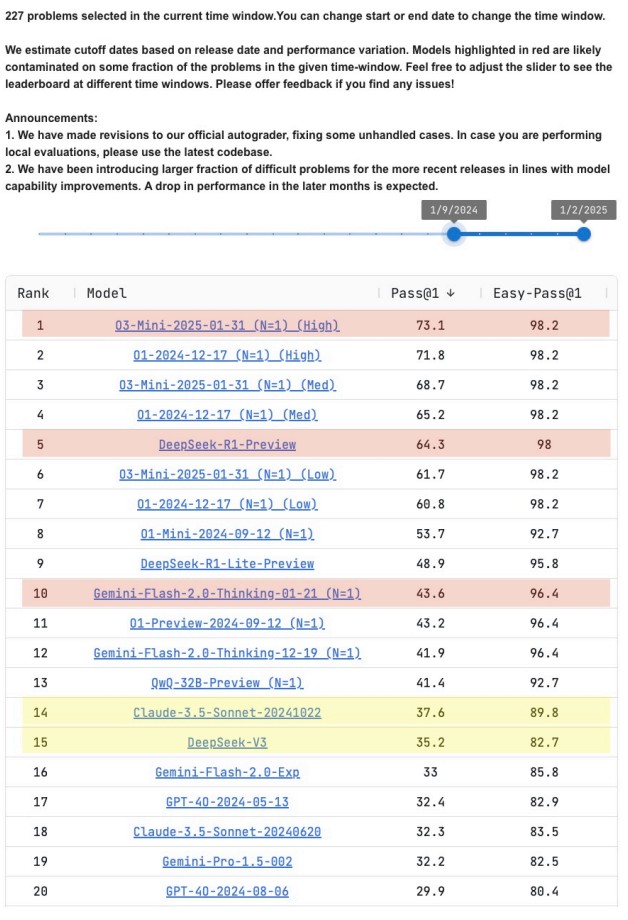

Figure 5: LiveCodeBench leaderboard in Feburary 2025, which we used to select the final 5 models used for our benchmarking. We took the top 3 reasoning models from unique developers, highlighted in red, and the top 2 chat models highlighted in yellow.

## B.1  Results on Hacked Subset

We report model performance on the "hacked subset" where humans found the counterexample in a extremely time-limit setting in Table 4.

Table 4: Performance of various strategies and models on the hacked subset of problems.

| Model | Zero-shot | Few-shot | w/ Correct | Agent w/o Demo | Agent w/ Demo |
|---|---|---|---|---|---|
| DeepSeek-V3 | 0.0 | 0.0 | 2.4 | 2.4 | 2.4 |
| Sonnet 3.5 | 4.8 | 4.8 | 2.4 | – | 2.4 |
| Flash 2.0 (Thinking) | 0.0 | 0.0 | 2.4 | 2.4 | 4.8 |
| DeepSeek-R1 | 2.4 | 2.4 | 2.4 | – | 2.4 |
| o3-mini (high) | 2.4 | 7.1 | 2.4 | 0.0 | 4.8 |

## B.2 Sample Length Analysis

| Sample component | Median Length |
|---|---|
| Problem Description | 1018 |
| Incorrect Submission | 1962 |
| Correct Solution | 1552 |

Table 5: We report median lengths (number of characters) of problem description and incorrect submission as these are fed as input to models. This affects the minimum input context length required and evaluation costs for our benchmark. We also report the median length of the correct solution for reference.

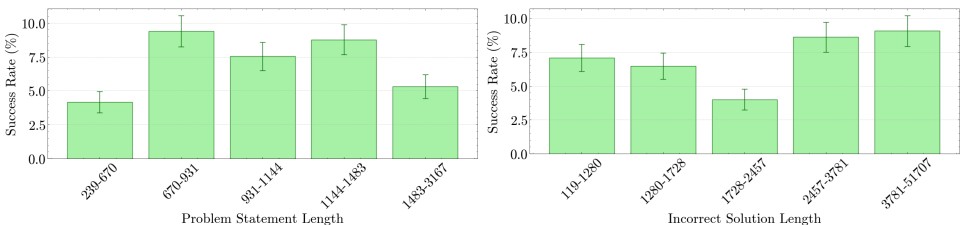

Figure 6: **Analysis of R1, o3-mini (high) success rate at creating counterexamples based on sample length, combining success across strategies like prompting, agent and programmatic search**. We find that model successes are not predictable based on problem statement length and incorrect solution length in characters.

# C Examples of Tasks: Samples from REFUTE

Although each incorrect submission requires individual reasoning to refute, we include a few examples showing three distinct kinds of bugs among the several present in our dataset in Figures 7, 8, and 9.

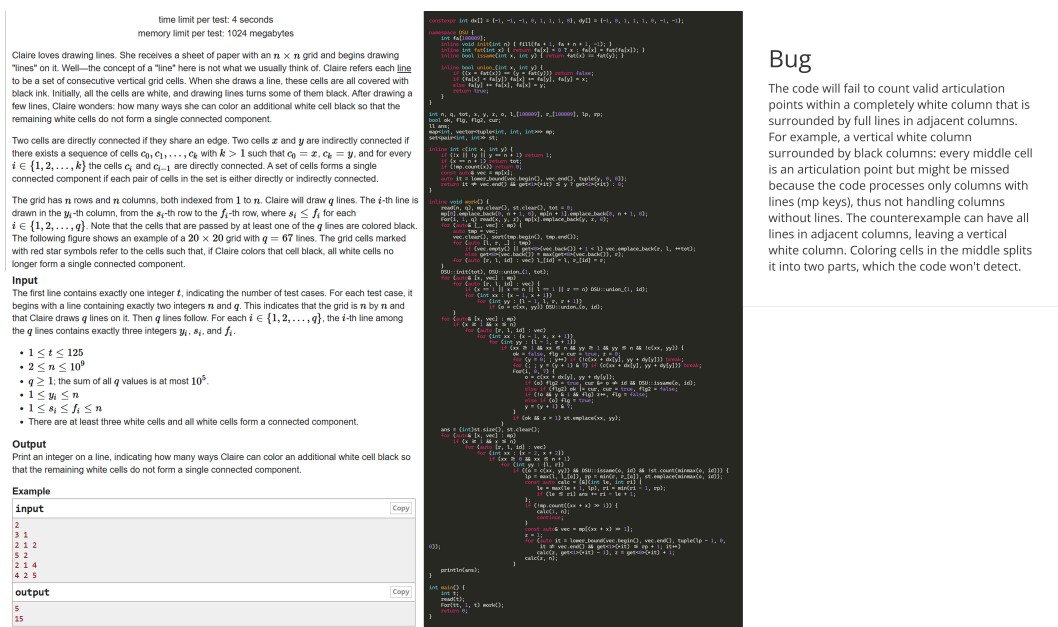

Figure 7: **REFUTE Sample 1.** The underlying approach in the code is flawed and the model must construct a grid which exposes it.

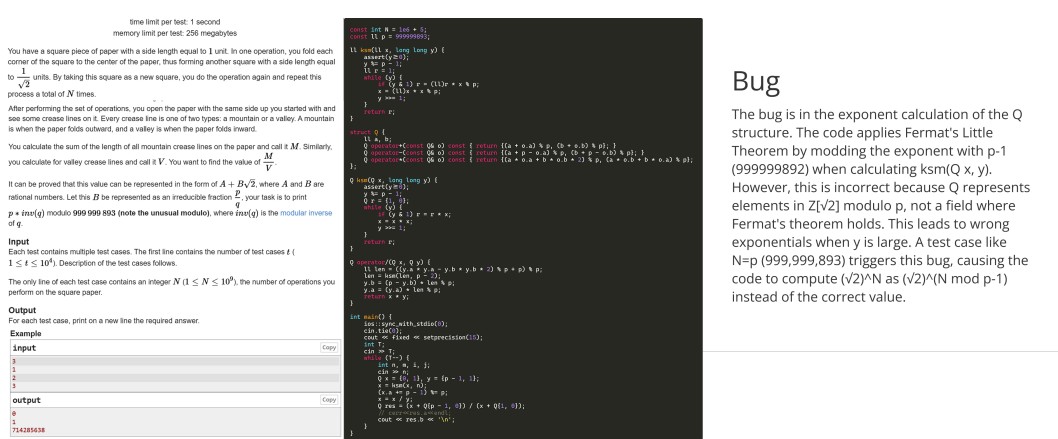

Figure 8: **REFUTE Sample 2.** The code demonstrates perfect understanding of the core solution, but makes an error while reporting it. The problem asks for the output to be reported in a certain format using exponents for the sake of easier evaluations, but the contestant made an error in assuming that using Fermat's theorem directly will work well in the given space. Only the part of the code responsible for reporting the answer needs to be fixed.

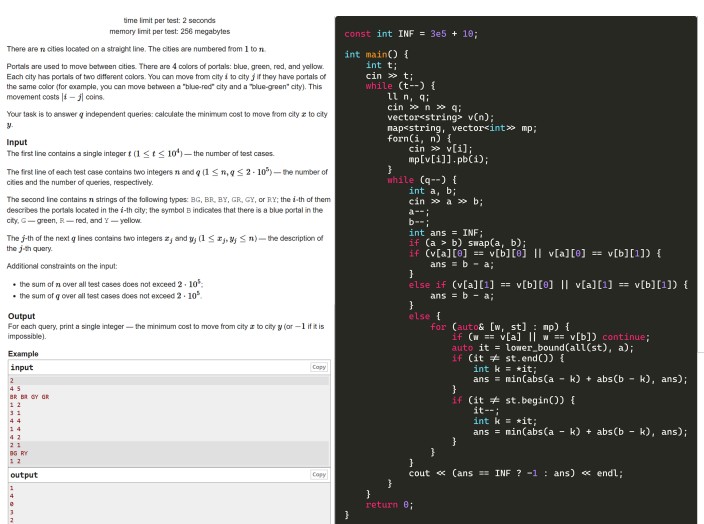

Figure 9: **REFUTE Sample 3.** The contestant simply underestimated what the maximum answer can be, and initialised a variable acting as a placeholder for 'Infinity' to be too small. Just doubling this value will fix the bug.

## D Future Extensions

A natural extension of our work is to broaden the scope of inverse benchmarks beyond algorithmic reasoning. While existing benchmarks like FrontierMath and HumanEval primarily assess problems-solving by requiring models to generate correct solutions that are directly matched against ground-truth, inverse benchmarks will require mechanisms to verify correctness of proposed claims over arbitrary instances. An example of such an interaction is provided in Figure 10.

Another promising avenue is to explore how the abstraction of an LLM attempting to falsify by generating code allows for hypothesis testing, data analysis, and simulations. In principle, code serves as a general medium that enables verification through diverse means – only limited by digital capabilities that models are fundamentally bound by already. This implies that leveraging code execution as a mechanism for falsification has broad applicability across diverse domains. As models improve, this paradigm may not only allow them to generate counterexamples but also to systematically explore patterns and behaviors, leading to more reliable scientific and mathematical discoveries.

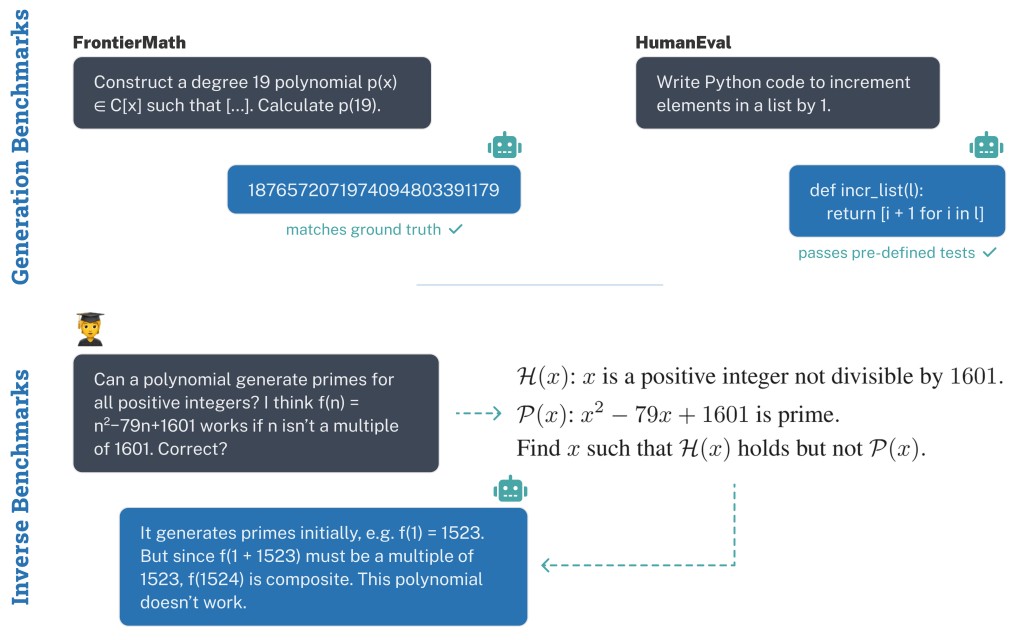

Figure 10: **Example of an inverse benchmark task in mathematics, contrasting with standard solution-generation benchmarks.** This illustrates how our general formulation of $\mathcal{H}(x)$ and $\mathcal{P}(x)$ (Section 3) extends to other domains, offering broad potential for future work.

# E    Limitations

We note several limitations of our current benchmark:

- **REFUTE benchmark is based on solutions in C++.** It could skew the benchmark in favor of models with higher C++ capability. However, this an inherent property stemming from the nature of competitive programming data source. While a Python-based version is possible, most real-world competition data is in C++. C++ is preferred due to its efficiency and suitability for strict runtime constraints. We highlight that the models we test are proficient in C++ given that their competitive programming capabilities are reported based on C++ solutions. This is also because there is abundant high-quality C++ code available online as training data for competition tasks.

- **REFUTE benchmark is only a proof of concept for how LLM falsification**. We highlight that REFUTE benchmark is not representative of general falsification capabilities. Our work is meant to demonstrate that: (1) Falsification capabilities can be evaluated as generating code that creates counterexamples which demonstrate a claim or solution is wrong. (2) As a first step, we chose a domain where automatic verification is relatively easy for novel counterexamples, and recent reasoning models already perform well at solution generation. We found that despite this, they are not good at generating counterexamples for incorrect solutions. More research will be required to create falsification benchmarks with better domain coverage and diversity

# F   Prompts

In this section, few-shot samples are denoted by `<examples>`. Similarly, long code snippets have been reduced to `[...]`. Full prompts with our samples and expert rationales are available in our code repository.

**Task Description Format**

The problem statement along with the incorrect code is presented to the model in this format. In all query formats described below, this is referred to as `<task_description>`.

```
## Statement

<problem_statement>

Time Limit: <time_limit>

Memory Limit: <memory_limit>

## Input Format
<input_format>

## Output Format
<output_format>

## Example Input
```
<example_inputs>
```

## Example Output
```
<example_outputs>
```

## Note
<optional_notes>

## Incorrect Code
```
<incorrect_code>
```
```

## F.1   Zero-Shot

```
System: You are an expert at finding errors in code. You will be given a
buggy code and the complete description of the problem it intends to solve.
Your job is to find a valid input in the expected format, satisfying all
input constraints, on which the code fails.

Write a program to print this failing test-case. Provide this to me in the
exact XML format I show below. Do not include anything other than your
thoughts and this program to print a failing test-case.

```
<reason>
[Your concise reasoning here]
</reason>
```

```
<action>
<name>print_fail_case</name>

[code to print failing test-case]

<lang>Python 3</lang>
</action>
```

**User:** You are now given a problem description and a buggy code. Help find a
failing test-case by providing a script in the exact XML format mentioned
earlier. Don't output anything else.

<task_description>

## F.2 Few-Shot

**System:** You are an expert at finding errors in code. You will be given a
buggy code and the complete description of the problem it intends to solve.
Your job is to find a valid input in the expected format, satisfying all
input constraints, on which the code fails.

Write a program to print this failing test-case. Provide this to me in the
exact XML format I show below. Do not include anything other than your
thoughts and this program to print a failing test-case.

```
<reason>
[Your concise reasoning here]
</reason>
<action>
<name>print_fail_case</name>

[code to print failing test-case]

<lang>Python 3</lang>
</action>
```

<examples>

**User:** You are now given a problem description and a buggy code. Help find a
failing test-case by providing a script in the exact XML format mentioned
earlier. Don't output anything else.

<task_description>

## F.3 Zero-Shot Agent

You are an expert at finding errors in code. You will be given a buggy code
and the complete description of the problem it intends to solve. Your job is
 to find a valid input in the expected format, satisfying all input
constraints, on which the code fails.

In your final submission, you need to provide a code to print this failing-
test case along with your reasoning to back it. If your generated test-case

doesn't match the input constraints and expected format, you will receive a VALIDATION_ERROR with relevant feedback. In such a case, you will have upto 5 chances to fix your submission. To make your submission, output an XML in the following format:
```
<reason>
[Your concise reasoning here]
</reason>

<action>
<name>print_fail_case</name>

[code to print failing test-case]

<lang>Python 3</lang>
</action>
```

You are also equipped with a code execution tool that you can use upto 10 times before your final submission. This will help you understand and narrow down to the failure case. You can execute any code you want with an arbitrary input. You will receive the output in response. Each code execution will be limited to 30 seconds. To use this tool, output an XML in the following format:
```
<reason>
[Your concise reasoning here]
</reason>

<action>
<name>run_code</name>

[source of the code you want to obtain output from]

<lang>[language of the source code -- one of 'Python 3' or 'C++ 23']</lang>
</action>

<action>
<name>input_print</name>

[code to print the input that will be passed to the code execution tool]

<lang>[language of the input printer -- one of 'Python 3' or 'C++ 23']</lang>
</action>
```

Your responses should ONLY be an XML in one of the two formats above. Thus, in an interaction, you will
- output an XML corresponding to the code-execution tool upto 10 times, and then
- output an XML for your final submission.

The interaction ends after you make a submission. **Use the code-execution tool generously** and only make a submission once you're certain of having found a failing test-case, or if you run out of your 10 attempts at the code-execution tool. You must use the code-execution tool atleast once. Utilise it generously to understand the code and verify your thoughts.

**User**: You are now given a problem description and a buggy code. Help me find a failing test-case using the code-execution tool and submission format provided above.

```
<task_description>
```

## F.4 ReAct Agent (With Sample Trajectory)

```
You are an expert at finding errors in code. You will be given a buggy code
and the complete description of the problem it intends to solve. Your job is
 to find a valid input in the expected format, satisfying all input
constraints, on which the code fails.

In your final submission, you need to provide a code to print this failing-
test case along with your reasoning to back it. If your generated test-case
doesn't match the input constraints and expected format, you will receive a
VALIDATION_ERROR with relevant feedback. In such a case, you will have upto
5 chances to fix your submission. To make your submission, output an XML in
the following format:
```
<reason>
[Your concise reasoning here]
</reason>

<action>
<name>print_fail_case</name>

[code to print failing test-case]

<lang>Python 3</lang>
</action>
```

You are also equipped with a code execution tool that you can use upto 10
times before your final submission. This will help you understand and narrow
 down to the failure case. You can execute any code you want with an
arbitrary input. You will receive the output in response. Each code
execution will be limited to 30 seconds. To use this tool, output an XML in
the following format:
```
<reason>
[Your concise reasoning here]
</reason>

<action>
<name>run_code</name>

[source of the code you want to obtain output from]

<lang>[language of the source code -- one of 'Python 3' or 'C++ 23']</lang>
</action>

<action>
<name>input_print</name>

[code to print the input that will be passed to the code execution tool]

<lang>[language of the input printer -- one of 'Python 3' or 'C++ 23']</lang
>
</action>
```

Your responses should ONLY be an XML in one of the two formats above. Thus,
in an interaction, you will
```

- output an XML corresponding to the code-execution tool upto 10 times, and then
- output an XML for your final submission.

The interaction ends after you make a submission. **Use the code-execution tool generously** and only make a submission once you're certain of having found a failing test-case, or if you run out of your 10 attempts at the code-execution tool. Your first output MUST use the code-execution tool.

Here's an example of a valid interaction. You'll be given an input:
<task_description>

You could respond with:
```
<reason>
The answer will not be 'Infinity' only when there is atleast one reaction each with requirements of the form (a, 0, 0), (0, b, 0), and (0, 0, c), where a, b, c > 0. This sets an independent upper bound on each of the three chemicals. The code seems to correctly handle this case.

In the other case, it appears to first sort the requirements array. It then continually adds the requirements (b, c) in a monotonic sequence which is increasing in b and decreasing in c. In another set, it stores the values of (b[i + 1] - 1 + c[i]) derived from consecutive entires in the earlier sequences, with the claim being that the optimal answer is always of this configuration. The claim seems to be correct, but I want to dig deeper into the implementation.

First, to verify my understanding, I have added an assertion to check whether prev(it) while pruning the monotonic sequence always refer to p1. I moved the original solver code to a separate function, and I call it repeatedly on randomly generated sequences. It this code runs out of time without throwing a runtime error, I will know that my understanding of that part was correct. Since I'm generating the inputs randomly inside the code, I don't need to provide a separate input generator.
</reason>

<action>
<name>run_code</name>

[...]

<lang>C++ 23</lang>
</action>

<action>
<name>input_print</name>

print("")

<lang>Python 3</lang>
</action>
```

You will then be given the input:
{"status": "TIME_LIMIT_EXCEEDED", "output": "Error while running source code: "}

You could then reply:
```
<reason>
Great, so I understood that part correctly. I find it suspicious that the code isn't checking if b[i + 1] > 0 before adding a corresponding option in
```

```
the st multiset. Because if b[i + 1] == 0 for some i in the monotonic
sequence ds, then I can't pick -1 as a valid amount of chemical b.

Let me add an assertion before every addition to st and run it on the sample
 first.
</reason>

<action>
<name>run_code</name>

[...]

<lang>C++ 23</lang>
</action>

<action>
<name>input_print</name>

print(4)
print("0 0 5")
print("0 5 0")
print("5 0 0")
print("6 7 8")

<lang>Python 3</lang>
</action>

```

You would then be given the input:
{"status": "RUNTIME_ERROR", "output": "Error while running source code:
Assertion failed: (*it).f > 0, file ./tmp\\code.cpp, line 79\r\n", "
return_code": 3}

You could then reply:
```
<reason>
Interesting, so even though I knew that code does provide the correct output
 on the samples, it fails the assertions that I put. So to exploit this, I
need to find a test-case where the wrong insertions to st do actually affect
 the output. I'll just plug the modified code (i.e. one that doesn't insert
invalid options into st) as a separate function, and then compare both the
given code and this modified code on random small inputs until I find one on
 which they differ.
</reason>

<action>
<name>run_code</name>

[...]

<lang>C++ 23</lang>
</action>

<action>
<name>input_print</name>

print("")

<lang>Python 3</lang>
</action>
```

```
You will then be given the input:
{"status": "OK", "output": "7\r\n0 0 7\r\n0 1 0\r\n0 9 2\r\n2 0 8\r\n6 5 8\r
\n7 0 0\r\n9 9 3\r\nfixed: 12, original: 13\r\n"}

You could choose to then reply with your final submission:
```
<reason>
Now that I've found an input on which I think the code is wrong (since it
output 13 instead of 12), I will use that as the submission to make via
print_fail_case.
</reason>

<action>
<name>print_fail_case</name>

s = """7\r\n0 0 7\r\n0 1 0\r\n0 9 2\r\n2 0 8\r\n6 5 8\r\n7 0 0\r\n9 9 3\r\n
"""
print(s)

<lang>Python 3</lang>
</action>
```

User: You are now given a problem description and a buggy code. Help me find
 a failing test-case using the code-execution tool and submission format
provided above.

<task_description>
```

## F.5 Few-Shot Random Search

```
System: You are an expert at finding errors in code. You will be given a
buggy code and the complete description of the problem it intends to solve.
Your job is to find a valid input in the expected format, satisfying all
input constraints, on which the code fails.

Write a program to print this failing test-case. Provide this to me in the
exact XML format I show below. Do not include anything other than your
thoughts and this program to print a failing test-case.

```
<reason>
[Your concise reasoning here]
</reason>
<action>
<name>print_fail_case</name>

[code to print failing test-case]

<lang>Python 3</lang>
</action>
```

<examples>

User: You are now given a problem description and a buggy code. Help find a
failing test-case by providing a script in the exact XML format mentioned
earlier. Don't output anything else.
```

```
<task_description>
```

## F.6 Zero-Shot (Oracle)

```
System: You are an expert at finding errors in code. You will be given the
complete description of a problem statement, along with a buggy code and
correct code to solve it. Your job is to find a valid input in the expected
format, satisfying all input constraints, on which the buggy code fails.

Write a program to print this failing test-case. Provide this to me in the
exact XML format I show below. Do not include anything other than your
thoughts and this program to print a failing test-case.

```
<reason>
[Your concise reasoning here]
</reason>
<action>
<name>print_fail_case</name>

[code to print failing test-case]

<lang>Python 3</lang>
</action>
```

User: You are now given a problem description, a buggy code, and a correct
code. Help find a failing test-case by providing a script in the exact XML
format mentioned earlier. Don't output anything else.

<task_description>

## Correct Code
```
<ground_truth_code>
```
```

## F.7 Random Search (Oracle)

```
System: You are an expert at testing code. You will be given the complete
description of a problem statement, along with a buggy code and correct code
 to solve it. You have to find a test-case where the buggy code fails. To do
 this, write a randomised test case generator script. I will then repeatedly
 compare the buggy code against the correct solution on the generator's
outputs until a failing test case is found.

Aim for diversity and coverage in the generated tests. Feel free to vary the
 range of all variables as needed while staying within problem constraints.

Provide the test-case generator to me in the exact XML format I show below.
Do not include anything else in your responses. Your code must be written in
 Python 3 or C++ 23.

```
<action>
<name>generate_tc</name>
```

```

[code to generate random test-cases]

<lang>[Python 3 | C++ 23]</lang>
</action>
```

<examples>

**User**: You are now given a problem description, a buggy code, and a correct
code. Write a testcase generator script in the exact XML format mentioned
earlier to find where the buggy code fails. Don't output anything else.

<task_description>

## Correct Code
```

<ground_truth_code>
```
```

