# OpenReview forum: "Can Language Models Falsify? Evaluating Algorithmic Reasoning with Counterexample Creation"
_colmweb.org/COLM/2025/Conference — COLM 2025_

### Official Review · Reviewer_sybC · 2025-05-10

**Rating:** 7
**Confidence:** 5
**Ethics Flag:** 1

**Summary:**

This paper introduces a novel benchmark, REFUTE, designed to evaluate a model's ability to falsify hypotheses. Based on code execution, the benchmark tests whether models can generate counterexamples for subtly incorrect solutions. The benchmark is well-curated and will be dynamically updated. It reveals several findings—for example, even OpenAI's o3-mini (high), when provided with code execution feedback, can generate counterexamples for fewer than 9% of the incorrect solutions in REFUTE. The authors also conduct further analyses and uncover interesting results, such as the observation that verification can sometimes be more challenging for models than solving the original problem correctly.

**Questions To Authors:**

1. Could the authors provide a discussion on how to improve the model’s ability to generate counterexamples based on the observed shortcomings? For example, what kinds of data or training methods might help enhance this capability?
2. In the analysis of factors that influence the model's ability to generate counterexamples, could the authors examine model performance across different algorithm types or problem topics? This could be a more meaningful attribute, as some algorithmic patterns might inherently make counterexample generation more difficult, while others might be easier. An analysis along these lines could offer deeper insights into the benchmark’s diagnostic power.

**Reasons To Accept:**

1. The research question addressed by the authors is valuable and timely. Even state-of-the-art models such as o3-mini (high), when equipped with code execution feedback, are able to generate counterexamples for <9% of incorrect solutions in REFUTE. This benchmark offers a relatively comprehensive evaluation of a model’s reasoning capabilities, particularly in falsification. It also has the potential to assess agentic capabilities when models are used in more autonomous or iterative settings.
2. REFUTE is a dynamically updating benchmark, which is especially important in the current era where data contamination is a growing concern. Continuous updates help ensure the benchmark remains challenging and free from training leakage.
3. The dataset is carefully constructed and yields insightful findings. For example, the results support the hypothesis that LLMs' difficulty in repairing their own incorrect code is often due to their limited ability to identify errors in the first place. In some cases, verification proves to be more challenging for models than solving the problem correctly.

**Reasons To Reject:**

1. While the authors note that most examples are in C++ because of its popularity in programming competitions, Python is undeniably the dominant language in the AI and machine learning community. From this perspective, the benchmark would be more broadly applicable and valuable if it included a greater proportion of Python examples.
2. The paper leaves some questions unanswered regarding when and why models can successfully generate counterexamples. Although the authors attempt an analysis across three attributes, they do not observe any clear patterns or trends. This may point to limitations in the benchmark’s design or coverage, particularly in its ability to disentangle the factors that influence counterexample generation success.

---

> ### Author Response · Authors · 2025-06-03
> **Rebuttal**
>
> We are glad you found our research question valuable and timely, dataset carefully constructed, and findings insightful. We appreciate the constructive feedback, and will address it below:
>
> >  the benchmark would be more broadly applicable and valuable if it included a greater proportion of Python examples.
>
> We acknowledge that the dataset primarily consists of C++ examples, reflecting the nature of competitive programming data sources. Although a Python-based benchmark is feasible, most competitive programming data remains predominantly in C++. C++ is preferred for its efficiency and suitability under strict runtime constraints, and models generally exhibit strong proficiency in this language. Evaluations by organizations such as OpenAI [1] also predominantly use C++-based Codeforces problems. Additionally, C++’s Standard Template Library (STL) conveniently provides built-in constructs suited for competitive programming. The abundance of high-quality training data available in C++ further biases model performance towards it. We will explicitly acknowledge the predominance of C++ as a limitation in our dataset.
>
> [1] OpenAI. "Competitive programming with large reasoning models." arXiv preprint arXiv:2502.06807 (2025).
>
> > Could the authors provide a discussion on how to improve the model’s ability to generate counterexamples based on the observed shortcomings?
>
> We now include a detailed analysis addressing when and why models succeed or fail in generating counterexamples in the common response in **Error Analysis** section.
>
> > In the analysis of factors influencing the model's ability to generate counterexamples, could the authors examine model performance across different algorithm types or problem topics?
>
> We provide some analysis in Figure 4. We will add the distribution of model performance across problem type tags in the revised draft.  We additionally conducted analysis over distribution of errors, which might be more relevant here and obtain the following results:
>
> | Factor | Model Success Rate | Dataset Rate |
> | --- | --- | --- |
> | **Initialization and bounds errors** | 56% | 32% |
> | **Implementation errors** | 12% | 20% |
> | **Edge case handling errors** | 3% | 20% |
> | **Incorrect approach** | 28% | 28% |
>
> We observe:
> - This distribution is notably skewed when compared to the underlying distribution of bug types in the dataset, which are more evenly balanced with no single dominating class, based on a manual analysis of a random sample of 100 problems.
> - Models are less likely to identify when the approach taken by the code is logically flawed, either irreparably or in very specific cases which can be separately handled. And they are much worse at spotting the latter.
> - On the other hand, they are more likely to find simpler issues with using wrong constants in code compared to what the input constraints would require, or other errors like off-by-one errors.
>
> We thank the reviewer for the suggestions and will be sure to include an explicit discussion of the error analysis, distribution of bug types and acknowledge the predominance of C++ as a limitation. We are glad this makes our paper stronger!

---

> > ### Comment · Reviewer_sybC · 2025-06-09
> >
> > Thank you for the detailed response. I still strongly encourage the authors to consider constructing more Python-based versions of the dataset, as it would greatly improve accessibility and community adoption—for example, I’d be personally interested in trying it out. Also, please make sure to include the promised analysis of model performance across different problem tags in the revised draft; I believe many readers will find it valuable. Thanks again for the thoughtful and thorough replies.

---

### Official Review · Reviewer_uUPk · 2025-05-11

**Rating:** 6
**Confidence:** 4
**Ethics Flag:** 1

**Summary:**

This work focuses on whether LLMs can find suitable counterexamples for fragile code appearing in programming problems. The authors collect problems from Codeforces, carefully clean the data, and design a new benchmark called REFUTE. After testing on state-of-the-art models, they find that existing LLMs may face greater challenges in error verification.

**Questions To Authors:**

The validation methodology adopted by the authors also deserves reconsideration. For example, if an LLM-generated counterexample leads to a program crash, it should be highly encouraged in the benchmark (as it exposes serious flaws in the solution); I would like to know how the benchmark handles some other special cases.

**Reasons To Accept:**

1. This work presents a novel perspective and addresses a critical problem: generating high-quality test cases has always been a challenge for LLMs. By designing this benchmark, the authors help draw the community’s attention to this issue and provide an effective testing scheme for future solutions.
2. This work conducts evaluations on state-of-the-art LLMs, ensuring both the challenge and the necessity of the proposed benchmark.

**Reasons To Reject:**

However, I believe there are several aspects where this work could be further improved:
First, the coverage and diversity of the benchmark are somewhat insufficient. The dataset is sourced from a few programming languages within Codeforces Division, and there is still a lack of sufficient evidence to prove that it is fully representative as a benchmark.
Second, the data quality of the benchmark is not fully demonstrated. Although the authors employed a large number of carefully designed automated methods to clean the data, credible evidence is still lacking to confirm that the benchmark data is of high quality.
Third, the authors should consider incorporating some reflective methods into the baselines to enhance the comprehensiveness of the benchmark [1, 2].
[1] Self-Refine: Iterative Refinement with Self-Feedback.
[2] Self-Consistency Improves Chain of Thought Reasoning in Language Models.

---

> ### Author Response · Authors · 2025-06-03
> **Rebuttal**
>
> We are glad you found our work a novel perspective on a critical problem, and our proposed evaluation effective. Below, we address the specific concerns raised:
>
> > First, the coverage and diversity of the benchmark are somewhat insufficient.
>
> We agree our benchmark is only a proof of concept for how LLM falsification capabilities can be measured, and not representative of general falsification capabilities. Our work is meant to demonstrate that:
>
> - 1. Falsification capabilities can be evaluated as generating code that creates counterexamples which demonstrate a claim or solution is wrong.
> - 2. As a first step, we chose a domain where automatic verification is relatively easy for novel counterexamples, and recent reasoning models already perform well at solution generation. We found that despite this, they are not good at generating counterexamples for incorrect solutions.
>
> We agree more research will be required to create falsification benchmarks with better domain coverage and diversity, and are happy to acknowledge this more explicitly in a new limitations section.
>
> >  Although the authors employed a large number of carefully designed automated methods to clean the data, credible evidence is still lacking to confirm that the benchmark data is of high quality.
>
> The data quality is enforced from the rules we used for data curation and filtering.
> - We only take official solutions on the codeforces website which are verified to be incorrect and fail some test case.
> - We highlight we had strict filtering of problems with a domain-expert, i.e. a grandmaster on Codeforces go through every problem and incorrect solution included in the benchmark to ensure its validity. The domain-expert found some initial solutions included conditional statements to make the code fail for some inputs, which are trivial samples for falsification. We explicitly excluded these.
>
> Overall, our dataset has **0% label noise**, in that *all* solutions are **verified** to be incorrect, and are **challenging** to falsify as they passed a *large* number of initial test cases.
>
> Additionally, the construction of our evaluation guarantees that any valid counterexample will be accepted by the evaluation procedure, and the model isn’t restricted in the ones in the ground truth. An ideal model can achieve 100% success rate, which is rare in LLM benchmarks [1].
>
> - [1] Vendrow, J., Vendrow, E., Beery, S., & Madry, A. (2025). Do Large Language Model Benchmarks Test Reliability?. *arXiv preprint arXiv:2502.03461*.
>
> > The authors should consider incorporating some reflective methods into the baselines to enhance the comprehensiveness of the benchmark [1, 2].
>
> We agree and will provide a comparison in the updated draft, we are currently running experiments on DeepseekV3 using self-reflection. We highlight that the reasoning models we tested are explicitly RL-trained with emergent reflection, and yet they performed poorly. It is officially recommended to keep prompts for reasoning models simple and direct, and recommendations for non-reasoning LLMs (prompting for chain of thought or reflection) may not apply: [https://platform.openai.com/docs/guides/reasoning-best-practices](https://platform.openai.com/docs/guides/reasoning-best-practices)
>
> We thank the reviewer for the suggestions and will be sure to include an explicit discussion of the data quality, and acknowledge the coverage as a limitation. We are glad this makes our paper stronger!

---

> > ### Comment · Reviewer_uUPk · 2025-06-08
> > **Responses to Rebuttal**
> >
> > Thank you to the authors for the detailed response. These replies have partially increased my confidence in the quality of the paper. For a benchmark, diversity and quality remain the most critical aspects. However, it seems that the authors are still unable to provide strong evidence to support these two points. Therefore, I would raise my score, but I still maintain my first and second concerns.

---

### Official Review · Reviewer_J67i · 2025-05-19

**Rating:** 7
**Confidence:** 3
**Ethics Flag:** 1

**Summary:**

This paper introduces a benchmark for finding counter-examples for incorrect solutions to programming questions. The benchmark is made up of Codeforces submissions and is filtered according to a list of criteria ensuring the benchmark requires algorithmic reasoning for the model to solve (avoiding brute force, syntax mistakes etc.). The paper evaluates a series of frontier models on the benchmarks across a number of different evaluation strategies (prompting, agentic interation, enforced random search). There is some analysis of failure modes.

**Questions To Authors:**

- I'm curious about the formula for problem selection. Did you try other coefficients for these terms / Did this make a difference in the types of problems sampled and the performance of different models?
- Could you articulate more clearly why the benchmark should not evaluate counter-examples that give OOT errors? Would identifying these gaps in the program not also display some reasonable understanding of the solution?
- Some of the results in Table 1 are counter-intuitive to me. Why do you think zero-shot performance is sometimes higher than few-shot or w/ correct answer? This would be easier to interpret with error bars. I have a similar question for the ReAct setting w/ demo.
- Could you include a couple samples of problems in your benchmark? It's difficult to assess the quality of the benchmark/how much it reflects the type of algorithmic reasoning it claims to without seeing a few examples.
- I'm also curious why these models (which are generally capable at programming and can solve many of the Codeforces questions) fail to write simple brute-force code? (this was identified as the key failure mode for random search). Could it be related to prompting strategy?

**Reasons To Accept:**

- There seems to be a great need in the community for benchmarks on different types of reasoning than simply solving math/algorithmic problems, especially for verification and falsification. I agree with the authors that this type of task helps delineate reasoning about the problem from simply parroting patterns from similar solutions (which models may be doing when generating solutions to these problems)
- This paper seems to be very thorough in the different methods of using the benchmark to evaluate the model (prompting, interaction, random search etc.). The interaction in particular is a great idea to assess whether a model is able to edit the code and interpret the difference in output (very valuable for co-operative programming type settings).
- The filtering process for selecting questions also seems to be quite thorough.

**Reasons To Reject:**

- I feel that the framing of this paper differs from the contribution. The framing in the introduction largely focuses on falsifying hypotheses in scientific discovery while the main contribution is to introduce a benchmark for finding counter-examples to programming problems, mostly in C++. It it not clear to me that these are the same kinds of reasoning and that being better at one would help the other. I don't see the value of the discussion related to scientific discovery.
- There are no error bars in the results tables. Was the evaluation conducted over multiple runs? It seems difficult to conclude how significant these results are and if models perform better on one evaluation strategy over others.
- The benchmark seems to massively oversample solutions in C++. Though I understand many Codeforces submissions are in C++, I wonder if this doesn't skew the benchmark in favor of models with higher C++ capability. I would feel more comfortable that the benchmark result actually reflected the ability of a model to find counter-examples if more languages were represented or some experiment was done to show this is not the case.
- Including an expert human baseline in Table 1 could help interpret that 9% result for o3-mini. I'm not sure how good/bad that is.
- In general, I think the contribution would be more valuable if the error analysis/identification of failure modes was more thorough. I don't feel that I've come away with a strong understanding of the specific failure modes of different models. There are no clear trends in the error analysis and it's surprising to me that models perform almost equally as well across problem difficulty and # test cases passed. Perhaps a more thorough investigation is needed.

---

> ### Author Response · Authors · 2025-06-03
> **Rebuttal**
>
> We are glad you found our research question of great need, and dataset curation thorough. We appreciate the constructive feedback, and will address it below:
>
> > The framing in the introduction largely focuses on falsifying hypotheses in scientific discovery while the main contribution is to introduce a benchmark for finding counter-examples to programming problems, mostly in C++. It it not clear to me that these are the same kinds of reasoning and that being better at one would help the other.
>
> Indeed, more research will be required to create an automatic evaluation for falsification of counterexamples to general scientific claims. Our work provides a proof of concept for algorithmic problem solving, where counterexamples are programmatically verifiable. We do think that our proposed abstraction of generating code that outputs a counterexample to a claim can generalize beyond coding scenarios, we characterize this in Section 3.1 and Section 3.2 which might be helpful. In Appendix B.3, we include a preliminary discussion of how the generated code could include entire experiments, as the agentic capabilities of LLMs improve going forward. In this domain where recent reasoning models have become quite good at generating solutions, we find they struggle at falsifying incorrect algorithmic solutions — implying being capable at generating solutions does not imply falsification capability, which might get overlooked given recent interest in the community about verification being *easier* than generation. We will include a reference to Section 3.1 and 3.2 to clarify our contribution, and highlight we tried to caveat this aspect even in the title. We are happy to be more up front about what our work achieves.
>
> > Was the evaluation conducted over multiple runs? It seems difficult to conclude how significant these results are
>
> We provide bootstrapped confidence intervals to quantify result variability. See below:
>
> | Model | Zero-shot | Few-shot | w/ Correct | Agent w/o Demo | Agent w/ Demo |
> | --- | --- | --- | --- | --- | --- |
> | DeepSeek-V3 | 2.4 [0.9, 4.2] | 2.7 [0.9, 4.5] | 3.7 [1.5, 5.8] | 3.7 [2.0, 5.9] | 3.1 [1.6, 5.0] |
> | Sonnet 3.5 | 4.6 [3.1, 6.9] | 3.7 [1.9, 5.9] | 2.2 [0.6, 3.7] | — | 3.0 [1.2, 5.1] |
> | Flash 2.0 (Thinking) | 0.9 [0.1, 2.1] | 2.1 [0.6, 3.7] | 2.5 [0.9, 4.3] | 1.8 [0.6, 3.4] | 2.5 [0.9, 4.2] |
> | DeepSeek-R1 | 5.8 [3.4, 8.5] | 5.2 [3.1, 7.9] | 4.6 [2.8, 7.1] |  | 6.5 [4.0, 9.2] |
> | o3-mini (high) | 8.6 [6.0, 11.6] | 8.9 [6.2, 11.9] | 9.3  [6.5, 12.2] | 6.8 [4.3, 9.5] | 8.6 [5.9, 11.6] |
>
> > The benchmark seems to massively oversample solutions in C++ ... I wonder if this doesn't skew the benchmark in favor of models with higher C++ capability.
>
> We acknowledge that the dataset is predominantly in C++, an inherent property stemming from the nature of competitive programming data source. While a Python-based version is possible, most real-world competition data is in C++. C++ is preferred due to its efficiency and suitability for strict runtime constraints. The models we test are proficient in C++ given that their competitive programming capabilities are reported based on C++ solutions [1]. This is also because there is abundant high-quality C++ code available online as training data for competition tasks. We will note the use of C++ as a limitation of our dataset.
>
> [1] OpenAI. "Competitive programming with large reasoning models." *arXiv preprint arXiv:2502.06807* (2025).
>
> > Including an expert human baseline in Table 1
>
> Note that all samples included in our dataset failed on human-generated test cases. In that sense, across problem setters, the human success rate was 100%. It is expensive to report a human baseline across samples here for a single human. However,  we did separately also report model performance on the “hacked subset” where humans found the counterexample in a extremely time-limit setting, and provide results below:
>
> | Model | Zero-shot | Few-shot | w/ Correct | Agent w/o Demo | Agent w/ Demo |
> | --- | --- | --- | --- | --- | --- |
> | DeepSeek-V3 | 0.0 | 0.0 | 2.4 | 2.4 | 2.4 |
> | Sonnet 3.5 | 4.8 | 4.8 | 2.4 | — | 2.4 |
> | Flash 2.0 (Thinking) | 0.0 | 0.0 | 2.4 | 2.4 | 4.8 |
> | DeepSeek-R1 | 2.4 | 2.4 | 2.4 | — | 2.4 |
> | o3-mini (high) | 2.4 | 7.1 | 2.4 | 0.0 | 4.8 |
>
> We discover the performance on the hacked subset is uniformly lower-- as these are the hardest subset of the problems. Every solution here passing the full array of test cases, including corner cases set by the problem setter.
>
> > I think the contribution would be more valuable if the error analysis/identification of failure modes was more thorough.
>
> We now include a detailed analysis addressing when and why models succeed or fail in generating counterexamples in the common response in **Error Analysis** section.

---

> > ### Author Response · Authors · 2025-06-03
> > **Questions**
> >
> > Questions:
> >
> > > I'm curious about the formula for problem selection.
> >
> > Great question. We’d first like to clarify the pipeline we use for selecting problems, that we elaborate in *Section 4.1 (Sourcing Problem Statements)*. This pipeline prioritises picking higher quality problems (evidenced by the division of the Codeforces contest that they appeared in) and then filters them for additional validity that we specify in the paper.
> >
> > The formula we mention in *Section 4.1 (Picking Incorrect Solutions)* is used to pick one incorrect code corresponding to each chosen problem, among the thousands that are available after running a filtering pipeline to find suitable code candidates. From this filtered set, we always want to prefer a hacked solution because they managed to pass all pre-made tests, and hence we assign that a large coefficient. From the other solutions, we want to pick one that passed several test cases before failing. But instead of picking one that maximises this value, we noticed that it was more fruitful to give a higher priority to those written by more experienced contestants. Their submissions did not have intentional artifacts designed for failure that we often found in submissions from lower rated contestants, and the bugs were more meaningfully subtle.
> >
> > > Could you articulate more clearly why the benchmark should not evaluate counter-examples that give OOT errors?
> >
> > Our goal is to evaluate semantic correctness aligning more closely with our motivation around falsification— whether a solution gives the wrong answer. Therefore, we focus on counterexamples that yield incorrect outputs. The category of performance-related failures like timeouts felt slightly orthogonal to our motivation as they reflect inefficiency rather than logical failure and trickier to assess consistently due to variable runtimes across systems.
> >
> > > Why do you think zero-shot performance is sometimes higher than few-shot or w/ correct answer?
> >
> > This is a known phenomenon in reasoning models — zero-shot performance can match or even exceed few-shot results (demonstrated in Deepseek-R1 tech report).
> >
> > > Include samples of problems from your benchmark
> >
> > We will add representative samples of benchmark problems in the appendix for clarity.
> >
> > > why these models (which are generally capable at programming and can solve many of the Codeforces questions) fail to write simple brute-force code?
> >
> > We address this point in the common response in **Error Analysis** section.
> >
> > > Could it be related to prompting strategy?
> >
> > Prompting variations did not significantly impact results. The models typically respond well to direct prompts, consistent with established prompting practices: [https://platform.openai.com/docs/guides/reasoning-best-practices](https://platform.openai.com/docs/guides/reasoning-best-practices).
> >
> > We thank the reviewer for the suggestions and will be sure to include the bootstrapped intervals, hacked subset experiments, problem statement selection justification, and acknowledge the coverage as a limitation. We are glad this makes our paper stronger!

---

> > ### Comment · Reviewer_J67i · 2025-06-03
> > **Follow-Up**
> >
> > Thank you for your complete response and efforts at improving the paper. A number of my concerns were addressed and I will await the Error Analysis comment. I have some concerns that remain:
> > 1. I agree with reviewer uUPk that the quality of the benchmark is not fully demonstrated in the paper. I understand the data curation process is thorough but it remains difficult to assess the quality and difficulty of the benchmark without seeing some representative examples. Including these during the discussion period would help gain a clearer understanding of what the benchmark is assessing exactly, the quality of samples and how difficult it is for the average human. It would also make any error analysis more concrete.
> > 2. Like reviewer LiKZ, I feel the core contribution of the paper does not align well with its current framing. I have carefully read the formalization in Sections 3.1 and 3.2, as well as Appendix B.3, but my concerns remain. While I appreciate the authors’ discussion, the paper still lacks convincing evidence that improvements in counter-example generation for programming problems would meaningfully advance falsification in scientific discovery. The link between the two remains speculative.
> >
> > That said, I do believe the proposed benchmark has potential value to the community and could serve as a useful first step toward studying falsification. To that end, I recommend that the framing in the abstract and introduction be revised to reflect this more modest scope. The connection to scientific discovery should be de-emphasized unless the authors can provide a more substantive justification or empirical support for it.

---

> > > ### Author Response · Authors · 2025-06-06
> > > **Follow-Up Rebuttal**
> > >
> > > We are glad that we could address your concerns! We hope that our analysis in the common comment helps provide further insights. Thank you for the follow up questions.
> > >
> > > > Including these during the discussion period would help gain a clearer understanding of what the benchmark is assessing exactly, the quality of samples and how difficult it is for the average human.
> > >
> > > Although each incorrect submission requires individual reasoning to refute, we have added a few examples showing three distinct kinds of bugs among the several: [REFUTE Samples](https://ibb.co/album/ZRcPkV). For each sample, we show the problem statement along with input and output constraints and demos to the model, along with the incorrect code (i.e. the first two columns). We added a third column in the images here with a brief annotation describing the bug, for the sake of clarity.
> > >
> > > - In the *Graph-Logic* sample, the underlying approach in the code is flawed and the model must construct a grid which exposes it.
> > > - On the other hand, in the *INF-initialisation* sample, the contestant simply underestimated what the maximum answer can be, and initialised a variable acting as a placeholder for ‘Infinity’ to be too small. Just doubling this value will fix the bug.
> > > - In the sample titled *Implementation*, the code demonstrates perfect understanding of the core solution, but makes an error while reporting it. The problem asks for the output to be reported in a certain format using exponents for the sake of easier evaluations, but the contestant made an error in assuming that using Fermat’s theorem directly will work well in the given space. Only the part of the code responsible for reporting the answer needs to be fixed.
> > >
> > > The model’s task in each case is only to find a counterexample. Notably, in the last task titled *Implementation*, both DeepSeek R1 and o3-mini guessed the issue to be with the core logic of the solution’s approach. But in the Oracle setting with access to correct code, both are successfully able to identify the bug and construct a testcase.
> > >
> > > > I recommend that the framing in the abstract and introduction be revised to reflect this more modest scope
> > >
> > > We agree that the paper will benefit from more clarity up front about the scope. To this end, we will revise the abstract and the introduction to better reflect this. We will also highlight further that the paper’s empirical contribution is regarding finding counterexamples for algorithmic reasoning.
> > >
> > > We thank the reviewer again for the thorough feedback and suggestions in helping improve our paper!

---

> > > > ### Comment · Reviewer_J67i · 2025-06-06
> > > > **Reply**
> > > >
> > > > Thank you for all of the work during the rebuttal. My concerns are addressed and I believe the paper should be accepted. Have edited my score to reflect this.

---

### Official Review · Reviewer_LiKZ · 2025-05-21

**Rating:** 7
**Confidence:** 4
**Ethics Flag:** 1

**Summary:**

This work presents a compelling and timely investigation into the capabilities of LLMs to engage in falsification through counterexample generation. It also introduces the REFUTE benchmark, aiming to assess LLMs' proficiency in identifying and challenging subtly incorrect algorithmic solutions. In terms of quality, the paper demonstrates a high level of technical soundness. The authors designed the REFUTE benchmark, ensuring that the tasks are both challenging and relevant to the evaluation of LLMs' reasoning abilities. The experimental setup is robust, with comprehensive evaluations that compare various LLMs' performances on the benchmark. The results are presented with clarity, and the authors provide thoughtful analyses that delve into the nuances of the models' successes and failures. As to the clarity, I found the paper well-written and well-structured, which is easy to understand and analyze. The introduction effectively sets the stage, articulating the motivation and significance of the study. The subsequent sections are organized logically, guiding the reader through the methodology, experiments, and discussions. In terms of originality, the paper makes a novel contribution by shifting the focus from LLMs' ability to generate correct solutions to their capacity to identify and challenge incorrect ones. The significance of this work is considerable. By highlighting the importance of falsification in algorithmic reasoning, the work addresses a critical aspect of AI development that has implications for the reliability and trustworthiness of LLMs. The REFUTE benchmark provides a valuable tool for the community, enabling more comprehensive evaluations of LLMs and potentially guiding the development of models with enhanced reasoning abilities.

**Questions To Authors:**

I would invite the authors to check the previous section where I shared some feedback about the weaknesses of the work, but I have an additional suggestion in order to strengthen their interesting work. I would suggest improving the empirical foundation and providing deeper insights into the challenges of counterexample generation, and consider conducting a comprehensive error analysis that categorizes the types of incorrect solutions and corresponding counterexamples within the REFUTE benchmark. By identifying patterns in the failures of LMs, such as specific algorithmic errors they struggle to detect or particular problem structures that hinder counterexample generation. I think this analysis would not only elucidate the specific reasoning deficiencies in existing LMs but also guide future research directions aimed at enhancing model robustness and self-reflection capabilities.

**Reasons To Accept:**

From my understanding, the novel and interesting contribution of this work is introducing the REFUTE benchmark, which shifts evaluation from traditional solution generation to the more cognitively demanding task of falsification through counterexample creation. This represents a clear step forward in how we assess model understanding and aligns closely with COLM's emphasis on deeper reasoning, self-reflection, and model robustness. In my opinion, the work is both conceptually novel and practically impactful, which addresses a blind spot in current benchmarks and proposes an evaluation setup that is grounded in formal semantics and executable verification, and I think it would be an interesting part of this work. The REFUTE benchmark is curated from real Codeforces problems with subtly incorrect solutions, ensuring that counterexample discovery is non-trivial and requires reasoning beyond surface-level understanding. From my view, the work's empirical design is very well, with evaluations across top-tier models using varied prompting strategies and agentic code execution. The finding that models capable of solving nearly 50% of problems fail to falsify over 90% of incorrect solutions is striking and reveals a deep limitation in current LM capabilities. I think this insight highlights a critical challenge in developing self-correcting AI systems, which is scientifically valuable. Beyond the methodology and analysis of this work, I see a strong technological impact. The benchmark fills a gap in the ecosystem of evaluation datasets, and I think it has the potential to become a community standard for measuring falsification. Its dynamic update strategy mitigates data leakage and ensures long-term relevance. Additionally, I think it would be important to mention that the work articulates a forward-looking vision: that models should be able not only to propose answers but also to challenge and reflect on them. This reframing of evaluation is intellectually bold and practically necessary as we move toward more autonomous and trustworthy AI.

**Reasons To Reject:**

While I think this work shares promising results and it has a valuable impact to be used in the community, I would like to share some concerns, and I hope the authors could find them useful. Please note that these concerns are to the best of my knowledge, and the main goal is to have a thorough discussion.

It seems to me that the paper frames counterexample generation as crucial for scientific discovery, yet the actual work is confined to a very specific domain of competitive programming problems. All experiments are conducted on REFUTE, a collection of 324 algorithms coding tasks drawn from Codeforces contests. While the authors hint at broader goals " with an eye towards subtly incorrect scientific claims ", I do not see any provided evidence that the approach generalizes beyond coding scenarios. I think it disconnect between the grand motivation (falsifying scientific hypotheses) and the limited score (debugging programming contest solutions) contributes feel narrow and conceptually overclaims. In its current form, the work comes across as a specialized benchmark for code debugging rather than a general advancement toward AI-driven falsification in science. The framing promises much more than what is actually achieved, which could mislead readers about the significance of the results.

---
The other thing that I would like to add is about the experimental section. From my understanding, there is no comparison to non-LLM approaches or human performance. A simple fuzzing or symbolic execution baseline could have provided a point of reference (e.g., can an automated programmatic method find counterexamples for any of these buggy solutions?). The authors claim to filter out trivial cases solvable by random search, yet they never quantify how a brute-force or heuristic search might perform on the remaining problems. I think this missing baseline makes it hard to gauge the true difficulty of REFUTE and how much LMs lag behind simpler methods.

Moreover, it seems that the evaluation is focusing only on success rates, which are extremely low, under 9% even for the best agent, without deeper analysis. Importantly, I think the paper would need to provide more insights into why the models fail. From what I have discovered in this work, providing the correct solution does not help much (only marginal gains in counterexample success), which highlights a fundamental gap between solving and falsifying. However, the authors do not investigate this gap beyond reporting the numbers. There are several things that I am going to mention. For instance, (1) Are the models failing because they can not comprehend the incorrect code logic, or because they can not generate complex inputs, or due to context length issues? (2) Are certain categories of bugs (e.g., greedy algorithm errors, off-by-one mistakes, etc.) more challenging than others? - I think knowing the details of them could be valuable for the community, and without understanding what the fundamental obstacles are, it would be a bit challenging moving forward.

---
The last comment that I have here is: The protocol allowing models up to 10 attempts with an " agentic " feedback loop is a reasonable approach inspired by how humans debug code. However, from my understanding, the paper does not justify key settings (why 10 attempts? Is this a random number? How sensitive are results to this limit?) and provides no ablation on the prompt strategies. It's also not clear to me whether the few-shot demonstrations were tuned for optimal performance or how those examples were chosen, which could affect success rates. The result table aggregates outcomes across all 324 tasks, but I think the paper could benefit from at least some breakdown (even if in appendix) - none is given, so it is a bit challenging to know if, say, a handful or trivial cases are being solved vs. everything else failing, or if all failures are uniformly distributed.

---

> ### Author Response · Authors · 2025-06-03
> **Rebuttal**
>
> We are glad you found our work impactful and experiments well designed. We appreciate the constructive feedback and address it below:
>
> > In its current form, the work comes across as a specialized benchmark for code debugging rather than a general advancement toward AI-driven falsification in science.
>
> Indeed, more research will be required to create an automatic evaluation for falsification of counterexamples to general scientific claims. Our work provides a proof of concept for algorithmic problem solving, where counterexamples are programatically verifiable. We do think that our proposed abstraction of generating code that outputs a counterexample to a claim can generalize beyond coding scenarios, we characterize this in Section 3.1 and Section 3.2 which might be helpful. In Appendix B.3, we include a preliminary discussion of how the generated code could include entire experiments, as the agentic capabilities of LLMs improve going forward. In this domain where recent reasoning models have become quite good at generating solutions, we find they struggle at falsifying incorrect algorithmic solutions — implying being capable at generating solutions does not imply falsification capability, which might get overlooked given recent interest in the community about verification being *easier* than generation. We will include a reference to Section 3.1 and 3.2 to clarify our contribution, and highlight we caveat this aspect even in the title. We are happy to be more up front about what our work achieves.
>
> > A simple fuzzing or symbolic execution baseline could have provided a point of reference (e.g., can an automated programmatic method find counterexamples for any of these buggy solutions?)
>
> Since the problems are stated in natural language, we are unsure how this would work. Could the reviewer please expand how fuzzing or symbolic execution systems handle natural language input? We agree such comparisons, if possible, would be quite interesting!
>
> > yet they never quantify how a brute-force or heuristic search might perform on the remaining problems.
>
> In Table 2, we showcase the *RandSearch* baseline where LLMs are prompted to generate a brute-force solution and randomly generated test cases given access to the problem and the incorrect solution. The obtained accuracies are low, with the best success rate being 8.3% from o3-mini (high). Note that this is different from the filtering step, where access to the incorrect solution was not provided, so any cases found this way were not a targeted falsification of the incorrect solution, and thus these samples were not included in the final dataset.
>
> >  provide more insights into why the models fail, consider conducting a comprehensive error analysis that categorizes the types of incorrect solutions and corresponding counterexamples within the REFUTE benchmark.
>
> We now include a detailed analysis addressing when and why models fail in the common response in **Error Analysis** section.
>
> > (1) Are the models failing because they can not comprehend the incorrect code logic, or because they can not generate complex inputs, or due to context length issues?
>
> We analysed the performance of models based on the problem statement and code length and provide the plot in Fig 6 (Appendix). We found that model successes are not predictable based on the length, and observe similarly poor performance on even the smallest code samples which are a few hundred characters. This indicates that context length issues are unlikely to be at play. Additionally, we note that we mitigate the difficulty in generating long counterexamples that follow a fixed patterns (which are commonly used to trigger bugs, and are the standard method for creating hacks on Codeforces) by letting the model output a script that prints the counterexample.
>
> > (2) Are certain categories of bugs (e.g., greedy algorithm errors, off-by-one mistakes, etc.) more challenging than others?
>
> We conducted an analysis over distribution of errors, and obtain the following results:
>
> | Factor | Model Success Rate | Dataset Rate |
> | --- | --- | --- |
> | **Initialization and bounds errors** | 56% | 32% |
> | **Implementation errors** | 12% | 20% |
> | **Edge case handling errors** | 3% | 20% |
> | **Incorrect approach** | 28% | 28% |
>
> We observe:
> - This distribution is notably skewed when compared to the underlying distribution of bug types in the dataset, which are more evenly balanced with no single dominating class, based on a manual analysis of a random sample of 50 problems.
> - Models are less likely to identify when the approach taken by the code is logically flawed, either irreparably or in very specific cases which can be separately handled. And they are much worse at spotting the latter.
> - On the other hand, they are more likely to find simpler issues with using wrong constants in code compared to what the input constraints would require, or other errors like off-by-one errors.

---

> > ### Author Response · Authors · 2025-06-03
> > **Rebuttal**
> >
> > > a handful or trivial cases are being solved vs. everything else failing, or if all failures are uniformly distributed.
> >
> > We do find that different LMs solve fairly different subsets of the problems. Most of the trivial cases were removed in our filtering pipeline, and thus there are no obvious trivial cases in REFUTE that all LLMs can solve.
> >
> > > identifying patterns in the failures of LMs, such as specific algorithmic errors they struggle to detect or particular problem structures that hinder counterexample generation.
> >
> > We analysed the ability of models to find bugs based on the category of the problem. To this end, we took the top 8 most frequent tags and compared the success rate of different strategies and models across them.
> >
> > - We found remarkably consistent trends with respect to the relative order of success rate across models. However, the trend depends on whether programmatic search or the other prompting methods are used.
> >
> > - For non programmatic search, overall accuracies across cateogories ranged from 6.2% (math) to 2.6% (trees and graphs). The best and worst performing categories stayed consistent across models and prompting methods.
> >
> > - On the other hand, programmatic search (RandSearch) more than doubled the success rate for problems that fell in the category of implementation-heavy, from 3.1% to 6.4%. The best perfoming category for RandSearch is constructive algorithms, whose success rate changed from 4.7% to 7.9%. The performance of in case of graph related problems fell to 1.6%, remaining the worst.
> >
> > - We’d want to highlight that the majority of the bugs in these graph related problems (which consistently proved to be the hardest across models and strategies) were related to the logic being incorrect and not just implementation issues, and thus the model would be required to construct a specifically structured graph to exploit the bug.
> >
> > Overall, we really appreciate the reviewer’s feedback which has greatly helped us improve our paper.

---

> > ### Comment · Reviewer_LiKZ · 2025-06-07
> > **Response from the reviewer**
> >
> > Thanks to the authors for addressing my concerns, and I hope you found my comments helpful to improve your work.
> >
> > As to the fuzzing or symbolic execution systems that you asked about, I would like to clarify that I did not mean to imply that fuzzing or symbolic execution systems would operate directly on natural language. Rather, my main point was to suggest that a baseline using these techniques could be constructed after grounding the problem statement and code into a formal or executable representation, as is already done in your pipeline when feeding incorrect solutions to LMs. For instance, your benchmark includes structured artifacts such as input constraints, code snippets, and validation logic, which could be used for symbolic execution or fuzz testing.
> >
> > Specifically, once the problem description and incorrect solution are parsed (as your system already does for prompting models and evaluating test cases), symbolic execution tools (e.g., KLEE or Z3) or fuzzers (e.g., AFL) could explore the input space for violating paths in the incorrect code. The symbolic execution engine can reason over paths that lead to divergence from expected outputs (perhaps compared against the correct reference solution, which you already provide), while fuzzers can be used to mutate inputs intelligently within the stated constraints. I was thinking maybe including such a baseline would contextualize your results and improve your claim that counterexample creation is a genuinely hard reasoning task for LMs, rather than merely a product of a large combinatorial input space.
> >
> > Thanks again for updating your manuscript. I am so happy to see that you have addressed my concerns, and I believe your work is strong enough to be accepted. I'd keep my score and wish you success

---

### Author Response · Authors · 2025-06-04
**Common Response: Error Analysis**

## Error Analysis

We analysed the failure modes by considering the outputs on a random set of 25 tasks, to understand where models fail. A domain expert (Codeforces Grandmaster) manually annotated the underlying bugs in the code, while comparing it against the models’ attempts.

- For 20% of the tasks, the model had an almost correct idea of the bug but couldn’t generate a testcase to exploit it.
- For 48% of the overall tasks, the model assumed that the bug is with underlying logical approach and completely missed that the issue was a minor implementation bug.
- The opposite happened only 8% of the samples, that they were thought to be an implementation issue when the backing logic was incorrect.
- The model correctly classified it as a logical issue with the approach or an implementation bug for 20% and 24% of the tasks respectively.

For the analysed incorrect submissions, we went ahead and checked the contestants’ performance during the contest, to get an idea of its difficulty for humans. We discovered that the contestants took a **median of 4 minutes combined to discover the bug, fix it, and resubmit an accepted solution.**

## Qualitative Analysis of Failures

We also performed a qualitative analysis to discover patterns in the failed attempts and common incorrect reasonings provided by the model, and present them below.

- We observed that models **frequently misallocate attention to parts of the code that are unlikely to contain the bug**. Specifically, rather than critiquing the high-level logic—a common first step among expert humans—the models tend to focus on subroutines or boilerplate components such as data structure templates or I/O code. These parts are typically less error-prone due to being standardized or reused. In contrast, humans often begin by assessing the overall algorithmic logic, refining attention to lower-level details only if needed. This discrepancy suggests that models may lack an effective abstraction for evaluating the structural correctness of a program and instead anchor on surface-level issues.
- Another common failure mode is **boundary condition analysis**. Human debuggers often stress-test edge cases by varying inputs to their extremal values within the constraint boundaries. Despite many of our benchmark instances being amenable to such bugs, models often failed to find corresponding counterexamples. This suggests limitations in the models’ ability to reason systematically over input domains and their boundaries.
- We also studied failure cases in the programmatic search settings (RandSearch and RandSearch Oracle). **Reasoning models (e.g., DeepSeek R1) often struggle to follow instructions requiring explicit use of randomized search.** They perform long chains of reasoning about possible counterexamples, and frequently produce deterministic code outputting a single test case. In contrast, chat models consistently adhered to the instruction of writing a randomized generator. This is one of the main reasons for the better performance of chat models like DeepSeek V3 in programmatic search compared to reasoning models like R1. While we attempted several prompt reformulations to encourage instruction-following in reasoning models, these efforts did not yield satisfactory improvements.
- A significant portion of counterexamples generated—particularly by DeepSeek R1—**failed format or constraint checks**, with almost 15% of them invalid in zero-shot and few-shot settings. However, **providing explicit feedback on *why* a counterexample failed validation and allowing for iterative correction (as in our agent setting) proved highly effective in improving this**, almost entirely removing validation issues. **Simply informing the model that a failure occurred, without specifying the reason, was insufficient**—even across multiple retry attempts. This indicates that grounded, interpretable feedback is helpful in improving model outputs.

## Distribution of Correctly Identified Bugs

We conduct an analysis of all successfully identified bugs, and obtain the following results:

| Factor | % of Successful Tasks | Dataset Rate |
| --- | --- | --- |
| **Initialization and bounds errors** | 56% | 32% |
| **Implementation errors** | 12% | 20% |
| **Edge case handling errors** | 3% | 20% |
| **Incorrect approach** | 28% | 28% |

We observe:
- This distribution is notably skewed when compared to the underlying distribution of bug types in the dataset, which are more evenly balanced with no single dominating class, based on a manual analysis of a random sample of 50 problems.
- Models are less likely to identify when the approach taken by the code is logically flawed, either irreparably or in very specific cases which can be separately handled. And they are much worse at spotting the latter.
- On the other hand, they are more likely to find simpler issues with using wrong constants in code compared to what the input constraints would require, or other errors like off-by-one errors.

---

### Decision · Program_Chairs · 2025-07-08

**Decision:**

Accept

**Comment:**

The paper presents a novel benchmark, REFUTE, for assessing the falsification capabilities of language models. The authors' approach leverages code execution to evaluate the ability of models to generate counterexamples for incorrect solutions, providing a comprehensive assessment of their reasoning capabilities. While the paper's scope is limited to competitive programming, the author addressed several concerns raised by the reviewers include quantitative and qualitative error analysis. Overall, I recommend accepting the paper due to its originality, and potential to inform future research, particularly in the areas of algorithmic reasoning and falsification.

Several potential areas of improvement: (1) empirically connect the ability to falsify to verification of outputs as correct or incorrect, (2) expanding the dataset to include Python problems, (3) analysis of benchmark performance as a function of test-time compute.